

# Bridging nano-optics and condensed matter formalisms in a unified description of inelastic scattering of relativistic electron beams

Hugo Lourenço-Martins[1,2]⋆, Axel Lubk[3] and Mathieu Kociak[4]

**1** Max Planck Institute for Biophysical Chemistry, 37077 Göttingen, Germany
**2** 4th Physical Institute, University of Göttingen, 37077 Göttingen, Germany
**3** Leibniz Institute for Solid State and Materials Research Dresden, Dresden, Germany
**4** Laboratoire de Physique des Solides, Université Paris-Saclay,
CNRS UMR 8502, F-91405, Orsay, France

⋆ hugo.lourenco-martins@uni-goettingen.de

## Abstract

In the last decades, the blossoming of experimental breakthroughs in the domain of electron energy loss spectroscopy (EELS) has triggered a variety of theoretical developments. Those have to deal with completely different situations, from atomically resolved phonon mapping to electron circular dichroism passing by surface plasmon mapping. All of them rely on very different physical approximations and have not yet been reconciled, despite early attempts to do so. As an effort in that direction, we report on the development of a scalar relativistic quantum electrodynamic (QED) approach of the inelastic scattering of fast electrons. This theory can be adapted to describe all modern EELS experiments, and under the relevant approximations, can be reduced to most of the last EELS theories. In that aim, we present in this paper the state of the art and the basics of scalar relativistic QED relevant to the electron inelastic scattering. We then give a clear relation between the two once antagonist descriptions of the EELS, the retarded dyadic Green function, usually applied to describe photonic excitations and the quasi-static mixed dynamic form factor (MDFF), more adapted to describe core electronic excitations of material. Using the photon propagator in a material, expressed in the relevant gauges, as a tool to understand the interaction between a fast electron and a material, we extend this relation to a newly defined quantity, the relativistic MDFF. The relation between the dyadic Green function and the relativistic MDFF does depend only on the photon propagator and not on the specifics of the particle (here, a fast electron) probing the target. Therefore, it can be adapted to any spectroscopy where a relation between the electromagnetic and electronic properties of a material is needed. We then use this theory to establish two important EELS-related equations. The first one relates the spatially resolved EELS to the imaginary part of the photon propagator and the incoming and outgoing electron beam wavefunction, synthesizing the most common theories developed for analyzing spatially resolved EELS experiments. The second one shows that the evolution of the electron beam density matrix is proportional to the mutual coherence tensor, proving that quite universally, the electromagnetic correlations in the target are imprinted in the coherence properties of the probing electron beam.

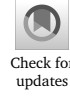

# Contents

# 1 Introduction

Electron energy loss spectroscopy (EELS) performed in a (scanning) transmission electron microscope ((S)TEM) analyzes the energy loss of fast electrons (with energy ranging typically from 30 to 300 keV) after their interaction with a target sample. It allows to probe a wide range of excitations in solids such as phonons [1], plasmons [2], excitons [3], and core electron-hole excitations [4] over a wide range of energies, typically from 40 meV to thousands of eV. Moreover, the versatility of the electron optical set-ups allows to achieve either high spatial resolution [5] (better than half an Ångström) or high momentum resolution [6,7] (smaller than $\mu m^{-1}$) or to probe directly the symmetry of the excitations [8]. This variety of experimental configurations is illustrated in Fig. 1.

A similar diversity is found in the theoretical descriptions of EELS experiments depending on how one treats the following aspects:

- The energy range of the probed excitations, usually dispatched in two classes, low-losses (less than tens of an eV) or core-losses (more than a hundred of eVs).

- The classical or quantum character of the beam electrons, corresponding to a description as either point charges or wave functions.

- The classical or quantum character of the target sample.

- The time-dependency of the electron wave function in the quantum formalism.

- The geometry of the experiment, especially whether the scattering events are analyzed in real or reciprocal space

- The retardation in the electron-sample interaction and in the fields propagating within the sample.

- The spatial overlap of the electron beam with the sample, e.g., the importance of bulk versus surface effects.

Since the general description (including all of the above cases) to EELS seems hardly viable, a plethora of different theories adapted to different sets of parameters have been developed. For example, the low-loss characterization of electromagnetic surface excitations such as surface plasmons is well interpretable assuming a classical character for both the electron beam and the target [9–11]. In such models the EEL cross-section is proportional to a well-known classical electromagnetic quantity, the (retarded) Green's dyad of the system [12]. The Green's dyad description is heavily used in nano-optics, necessitating an accurate description of electromagnetic field propagation within complex geometries of dielectric media. On the opposite side, core-electron excitations are usually described in a Fermi's Golden rule approach rooted in a quantum treatment of the inelastic scattering processes between the beam electron and the target degrees of freedom [13]. This approach can be generalized by employing the mixed dynamic form factor (MDFF) formalism introduced by Kohl and Rose [14] which typically neglects retardation as well as many-body effects in the target beyond the mean field level, when applied to core losses.

Both the Green's dyad and the MDFF descriptions have been extended, thereby approaching each other. For example, by using a fluctuation-dissipation approach the Green's dyad formalism has been adapted in order to take into account the quantum character of the electron [11]. Many-body effects, on the other hand, have been absorbed in the MDFF to describe low-loss volume excitations, such as plasmons (on the random phase approximation level) [15,16] or excitons (employing Bethe-Salpeter approximations) [17] in condensed matter systems.

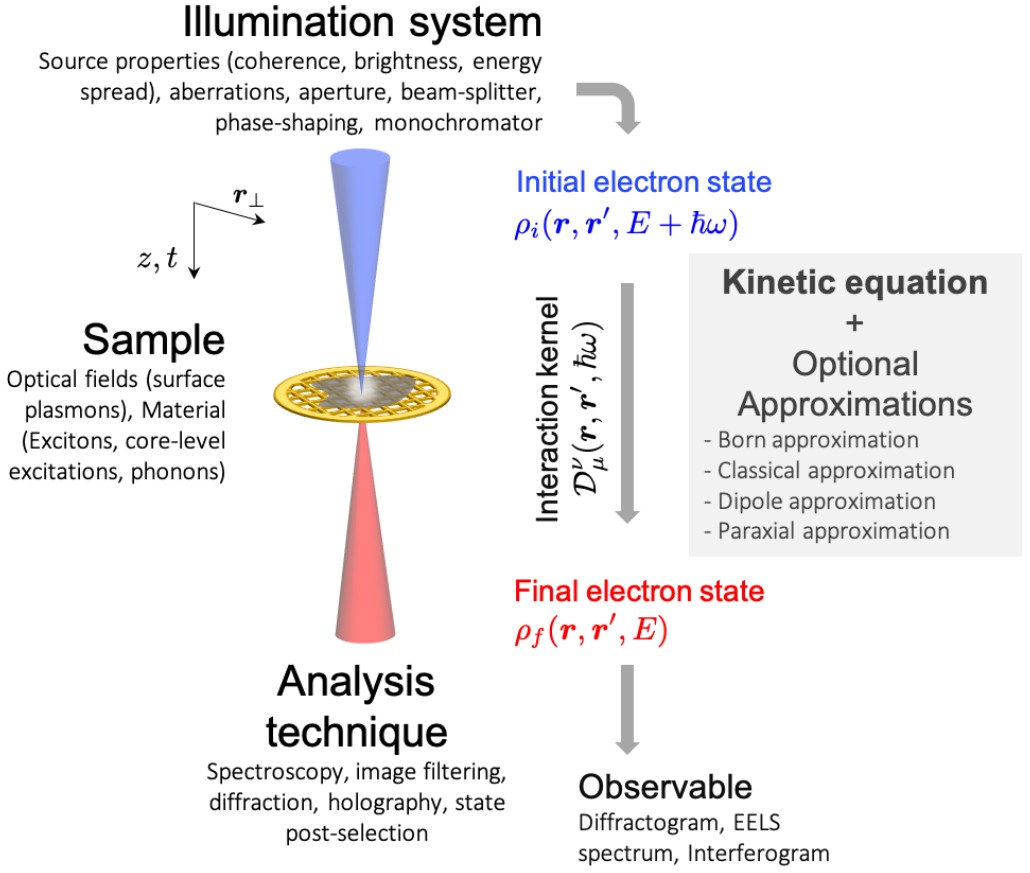

Figure 1: Schematics representing the general experimental conditions considered in this paper. A fast electron described by the density matrix $\rho_i$ is scattered by a target to the final density matrix $\rho_f$. TEM analysis techniques (EELS, diffraction, holography, ...) allow to extract information contained in this final electron state. The gist of our work is to derive the general kinetic equation connecting the initial to the final electron beam state through an interaction kernel embedding all the physical details (classical, relativistic or quantum) of the target.

The general relation between the two approaches, however, has still not been established, which is unsatisfactory not only from an intellectual point of view but also concerning the interpretation of EEL spectra in different experimental settings. Indeed, it raises unnecessary conceptual walls between alternative descriptions of the electron beam, of the target excitations, and of the electron to target interaction. For example, it has been demonstrated [18] that the so-called magic angle, at which the dependence of the core-loss electron scattering on the orientation of an anisotropic sample is canceled, strongly depends on the retarded character of the electron to target interaction, which had been considered as negligible in core-loss investigations before. Also, the description of the effect of interferences between inelastically scattered electrons, otherwise speaking, coherence effects, has been long discussed in terms of MDFF in the framework of inelastic holography of bulk plasmons [19] or EELS atomic resolution mapping [20–22], while coherence effects in surface plasmon excitations relied on Green's Dyad approaches [23]. Even more, electron magnetic circular dichroism (EMCD) [24] and phase-shaped EELS applied to plasmonics [8,23] relies physically on the same ingredients - electron phase manipulation to mimic the action of an X-ray or optical photon - yet are

described within totally different frameworks.

It is therefore not surprising that early on, researchers have sought for a comprehensive description of EELS in the (S)TEM, or at least for a bridge between different approximations. For example, Ritchie and Howie [25] could explain how the interferences of inelastically scattered electrons are washed out by integrating over all scattering angles rendering the quantum and classical descriptions of the electrons essentially equivalent for most of the experimentally relevant low-loss STEM-EELS cases. In order to model dynamical scattering effects in diffraction, Dudarev, Peng and Whelan [26] drew a direct link between the density matrix of the electron beam and the MDFF in the quasi-static case. Later on, Schattschneider et al. [15] applied a similar approach to relate EELS to the MDFF, and to establish a clear link between reciprocal and real-space representations of the electron-target interaction. Schattschneider and Lichte [19] later used the MDFF formalism to properly describe coherence effects between electrons in the quasi-static limit, following the pioneering work of Kohl and Rose [14]. Later, García de Abajo proposed a fully relativistic description of low-loss EELS experiments, where the quantum nature of the probe electrons could be taken into account [11, 23]. Nevertheless, no universal description of EELS in a (S)TEM has been provided, which implies that the relation between the different approximations remained somewhat in the dark.

The present work is an effort towards the goal of giving such a description. By extending and bridging several key works [11, 14, 15, 26] we provide a synthetic description of EELS in a (S)TEM. This description is valid for any sort of classical or quantum description of the electrons, arbitrary equilibrium description of the target object, any sort of excitations (low-loss and core-loss, surface and bulk) and using retarded or non-retarded approaches alike. Incidentally, we are formally establishing the link between the Dyad and the MDDF approach, extending the latter to the retarded case, therefore widening the applicability of our work to other spectroscopies not necessarily involving electrons.

Working out such a theory is challenging because both quantum and relativistic effects need to be taken into account. As a consequence, the problem of computing the complete beam electron-target interaction cannot be solved in a closed form and different approximation schemes have to be employed, notably diagrammatic perturbation techniques. Accordingly, we won't consider effects connected to the finite temporal length of electron wavepackets [27–30]. In other words, the wavefunctions encountered throughout this paper do not represent quantum wavepackets but rather electron beams in a steady-state illumination, in strict analogy with photon wave optics [31]. With this in mind, we can synthesise the electron energy-loss processes studied in this paper with a diagrammatic perturbation language roughly schematized in Fig. 2.

Here, the inital and final beam electron state are represented by the wave functions $\psi_{i,f}(\mathbf{r}, t)$ and the target by the wave functions $\xi_{i,f}(\mathbf{r}, t)$. (Virtual) photons, indicated by wiggly lines, of arbitrary numbers and orderings are exchanged in the course of interaction. In analysing the diagram, we first note that relativistic effects can emerge from both the probe and the target [32]. Indeed, beam acceleration voltages in a TEM typically range from 80 to 300kV leading to electron velocities $v$ between 0.5 and 0.78$c$, and variations of the Lorentz factor

$$\gamma = \frac{1}{\sqrt{1 - v^2/c^2}} \tag{1}$$

between 1.16 to 1.59 [33]. Therefore, TEM electrons are relativistic, which translates into (A) a modification of the kinetic properties of the electron such as re-normalized masses [34], (B) retardation effects in the electromagnetic interaction [35], (C) Cherenkov losses [6, 36, 37], and (D) sizeable current interaction effects. While (A) and (D) directly scale with the velocity of the electron beam, the retardation effects in the interaction cannot be neglected anymore when the length scale $L$ of the charge density fluctuations associated with an excitation of

**wave diagram**

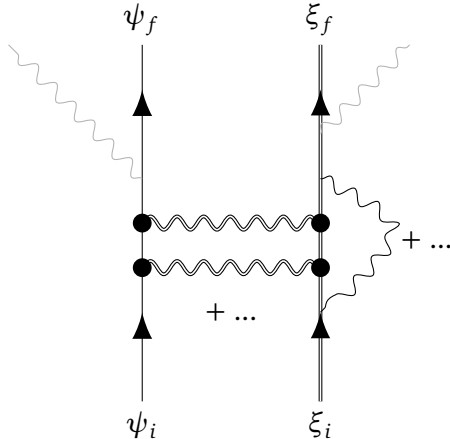

Figure 2: Diagram representation of the electron energy loss process: a beam electron described by a wavefunction $\psi(r,t)$ interacts with a target represented by a (many-body) wave function $\xi(r,t)$ by exchanging virtual photons. The target wave function and photon propagators are generally dressed (renormalized) by interacting with the various degrees of freedoms of the target (indicated by double lines). Processes involving the emission of photons (represented by gray lines), such as Bremsstrahlung, are not considered in this paper.

energy $\omega$ in the target become important, i. e. when $\omega L/c > 1$ (see [32] and references therein). This situation typically occurs in plasmonics which leads to a frequency red-shift, a loss of spatial coherence or even to mode splitting [38].

The large energy of the beam electrons, however, also renders spin-orbit coupling effects in the scattering process itself negligible [39, 40], hence a full quantum relativistic modeling of the electron beam is not required. Indeed, approaches to EELS or electron diffraction based on the Dirac equations have been developed [34, 41–44] and give results comparable to what the Klein-Gordon equations do [26, 45–47].

In what follows we deliberately focus on EELS including all possible electron energy-loss mechanisms. However, we do not explicitly compute (secondary) scattering events involving the emission of photons (indicated gray in Fig. 2), should it be due to Cerenkov, Bremsstrahlung or cathodoluminescence for example, in order to keep the length of the paper at bay. However the perturbation technique used throughout is well suited to also describe these events and may be easily extended. Most importantly, we will restrict the inelastic interaction to the first order Born level, which, however, does not exclude to consider multiple stacked first order Born events as in Sec. 6.

The basis for our considerations is the standard diagram perturbation technique as discussed in, e.g., [48]. Accordingly, in order to properly take into account many body effects in the target (e.g., inelastic interaction with the electron gas at the Fermi level) and partial coherence in the beam (e.g., as produced via inelastic interaction) a generalization of the wave diagram in Fig. 2 to a density operator diagram (i.e., in the language of second quantization) techniques is in order (Fig. 3). The corresponding diagram is obtained by taking the tensor product of the wave diagram (as indicated) and identifying the initial and final states of the target on both sides, which follows from tracing over all target degrees of freedom.

Here, the gray loop generally contains all possible diagrams up to infinite order, i.e., the exchange of (dressed) virtual photon lines in the target indicated in Fig. 2. It is referred

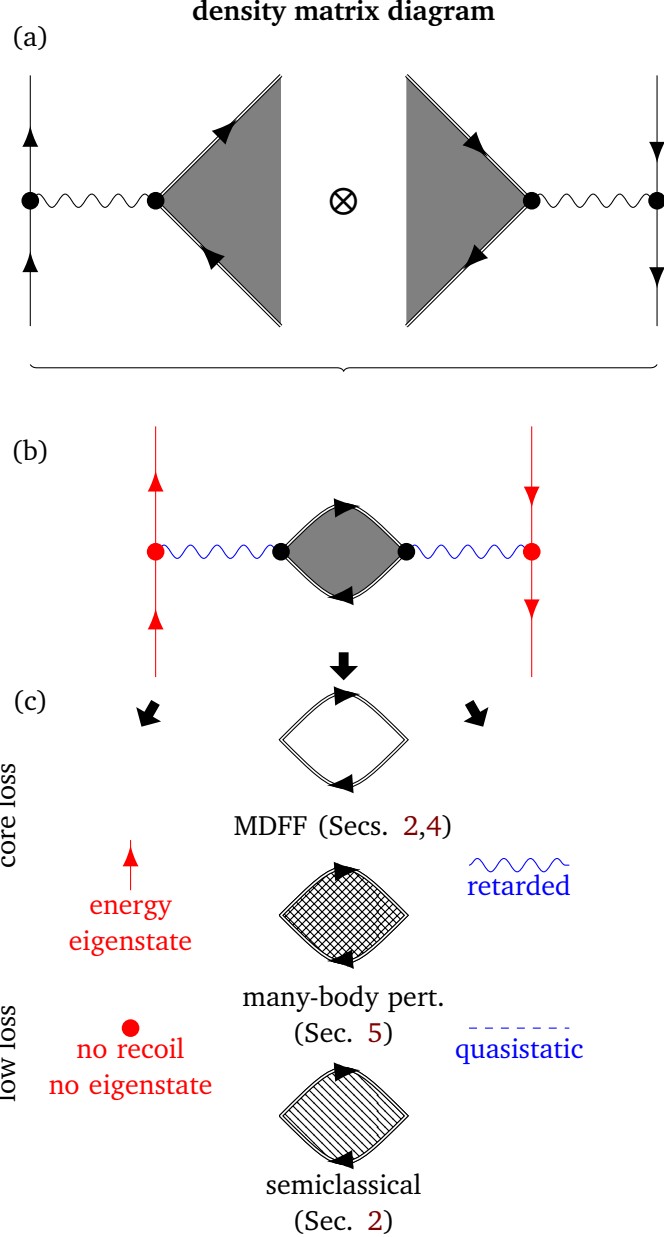

Figure 3: Density (matrix) diagram representations of the inelastic scattering process including various approximations: as the initial and final sample state are the same on the left and right hand side of (a), the target side of the tensor product can be connected forming a loop, i.e. a 4-point correlation (response) function or (generalized) two particle Green's function (b). All ingredients of the resulting diagram are subject to various approximations (c), all of which treated in the paper.

to as 4-point correlation function, two particle function, polarization propagator or generalized Green's function in the literature. It represents the fundamental object encapsulating the physics of the excitations of the target. It is well suited to encompass many-body interaction effects (indicated gray) in the target, which particularly affect the low-loss regime. In this article we will not elaborate on the various sophisticated strategies which have been developed to compute the 4-point correlation function (typically relying on some infinite series expansions

in the interaction). We will rather discuss the various descriptions of the beam electron, the (virtual) photon exchange and the ramifications of relativity.

Indeed, all relevant descriptions of EELS may be identified with certain approximations to the general diagram in Fig. 2. In particular neglecting many-body interactions (beyond the mean field level) in the target corresponds to the MDFF used in core-loss computations, whereas the Green's dyad is a semiclassical version of the photon propagators including the correlation function. Accordingly, Fig. 3 is well suited to provide an overview of what is treated in this work. In Sec. 2 we give a more detailed account of the state of the art of EELS in terms of experimental setups and pertinent theoretical descriptions, corresponding to various combinations of approximations indicated in Fig. 3. In Sec. 3, we introduce the various conventions and notations used throughout this paper. We also recall some results on free-space photon propagators and their expression in different gauges.

In Sec. 4 we review electron scattering within the framework of scalar relativistic quantum electrodynamic (QED). The main conclusion of this part is that in addition to the possibly retarded Coulomb interaction (i.e., a charge-charge coupling term) also a magnetic interaction (i.e., a current-current interaction) needs to be considered because of the relativistic velocities of the beam electrons. In Sec. 5 we focus on the 2-particle function (see Fig. 3). For completeness and pedagogy, we first re-derive the expression for the MDFF and connect it to the screened interaction. This way, we draw a parallel between the quasi-static formalism of nano-optics with the one of core-loss spectroscopy. We then focus on the retarded case, where we heavily rely on the fact that the interaction can be modeled in terms of correlation functions and photon propagators (see Figure 3). Using a pedestrian approach of the QED, we calculate the exact photon propagator in the presence of a polarizable material, taking special care of the gauge choices. In complete analogy to what has been done with the MDFF in the quasi-static case, we then connect this photonic kernel to the charge and current density correlation functions of the scatterer. This makes it possible to introduce a fully retarded definition of the MDFF as the spatial Fourier transform of the 4-susceptibility of the target. By doing that, we introduce relations that bridge the world of solid-state physics and optics and that can be applied well beyond the case of electron energy loss spectroscopy.

In Sec. 6 we then consider the electron probe part in detail. For reasons exposed earlier, we model the electron propagation using the Schrödinger equation with a semi-relativistic correction, i.e. mass renormalization [47]. Following the seminal demonstration of Dudarev and collaborators [26], we employ the Bethe-Salpeter equation in the ladder approximation in order to calculate the electron density matrix propagation equation. We then consider separately the quasi-static and the retarded interaction, which give the kinetic equation of Dudarev et al. and its retarded counterpart, respectively.

Sec. 7 is dedicated to the contextualization of our developments and proposal of different applications. Particularly, by taking the appropriate limits, we demonstrate that our equations encompass all previously obtained results.

As important applications of the formalism developed in this paper, we provide three important formulas. For the sake of pedagogy, we present them here, but they will be reproduced, derived and discussed in length in this manuscript. The first is a concise and universal formula for interpreting the most commonly performed EELS experiments and reads:

$$\Gamma(\omega) \sim \sum_f \int d\boldsymbol{x}\, d\boldsymbol{x}'\, \psi_f^*(\boldsymbol{x}) \psi_f(\boldsymbol{x}') \hat{C}(\boldsymbol{x}, \boldsymbol{x}', \omega) \psi_i(\boldsymbol{x}) \psi_i^*(\boldsymbol{x}') \delta(\epsilon_i - \epsilon_f - \omega). \tag{2}$$

This formula relates directly, within the framework of Fermi's golden rule (first order Born approximation), the loss probability per unit time $\Gamma$ (i.e., the EEL spectrum) to the incoming electron wavefunction ($\psi_i$) at points $\boldsymbol{x}$ and $\boldsymbol{x}'$, the related scattered electron wavefunctions

Table 1: Table synthesizing the different optical-condensed matter analogue quantities and equations as well as the relations between them. The Mutual coherence tensor $\Lambda_{ij}$ denotes the central object of nano-optics from which any other important quantity - such as the electromagnetic local density of states (EMLDOS) or the cross-density of states (CDOS) - can be deduced, as done in Sec. 3.2.

| Regime | Quantity | Optical description | Connection | Condensed matter description |
|---|---|---|---|---|
| Quasi-static | Propagator | Screened Greens function $\mathcal{W}(\boldsymbol{r}, \boldsymbol{r}', \omega)$, (10) | (61) | Mixed dynamic form factor (MDFF) $S(\boldsymbol{k}, \boldsymbol{k}', \omega)$, (15) |
| Quasi-static | Correlation function | Potential correlation function $\langle A_0(\boldsymbol{r}', \omega) A_0(\boldsymbol{r}', \omega)\rangle$, (148) | (56) | Charge-Charge correlation function $\chi(\boldsymbol{k}', \boldsymbol{k}, \omega)$, (54) |
| Quasi-static | Kinetic equation | (145) | (61) | (14) |
| Retarded | Propagator | Retarded screened interaction $\mathcal{D}_{\mu\nu}(\boldsymbol{r}, \boldsymbol{r}', \omega)$, (41) (148) | (45) | Relativistic Mixed dynamic form factor $S_{\mu\nu}(\boldsymbol{k}, \boldsymbol{k}', \omega)$, (43) |
| Retarded | Correlation function | Mutual coherence tensor $\Lambda_{i,j}(\boldsymbol{r}, \boldsymbol{r}', \omega)$, (24) | (97) | 4-Current correlation functions $\chi_{\mu\nu}(\boldsymbol{k}', \boldsymbol{k}, \omega)$, (43) (69) |
| Retarded | Kinetic equation | (151) (154) | (45) | (152) |

$\psi_f$, and the correlation function of the target $\hat{C}(\boldsymbol{x}, \boldsymbol{x}')$. The later can be calculated for the relevant charge excitations such as bulk or surface phonons, plasmons, core-electrons, etc., in either the quasistatic limit or retarded case, as discussed previously (see Fig. 3). It is therefore the most general and synthetic formula for computing EELS. The second formula reads:

$$\rho_f\left(\boldsymbol{r}_\perp, \boldsymbol{r}_\perp'\right) \sim \frac{1}{\omega} \Lambda_{zz}(\boldsymbol{r}_\perp, \boldsymbol{r}_\perp', q_z, -q_z, \omega) \, \rho_i\left(\boldsymbol{r}_\perp, \boldsymbol{r}_\perp'\right), \tag{3}$$

where $\boldsymbol{r} = (\boldsymbol{r}_\perp, z)$ respectively denotes the transverse (orthogonal to the electron beam trajectory) and longitudinal (along the electron beam trajectory) coordinates. Here, the *mutual coherence tensor* [49,50] $\Lambda_{zz}(\boldsymbol{r}_\perp, \boldsymbol{r}_\perp', q_z, -q_z, \omega)$ relates the transition from initial to final density matrices, $\rho_i\left(\boldsymbol{r}_\perp, \boldsymbol{r}_\perp'\right)$ and $\rho_f\left(\boldsymbol{r}_\perp, \boldsymbol{r}_\perp'\right)$, of the probe electron by an energy loss process of energy $\hbar\omega$ and characteristic momentum $q_z = \omega/v$. The $zz$ indices hint at the tensorial character of $\Lambda$, which we can typically neglect in EELS because of the small scattering angles of the fast beam electrons around the $z$-direction (paraxial approximation). This generalization of the quasi-static work of Kohl and Rose [14] shows that quite universally, the electromagnetic correlations of the target are imprinted in the coherence properties of the electron beam, and makes straightforward the interpretation of holographic or phase-dependent experiments regardless of how the target is described. The third formula reads:

$$S_{\mu\nu}(\boldsymbol{k}, \boldsymbol{k}', \omega) = \text{Im}\{-\chi_{\mu\nu}\left(\boldsymbol{k}, \boldsymbol{k}', \omega\right)\} = 2\pi \sum_f \langle i|\mathrm{j}_\mu(\boldsymbol{k})|f\rangle \langle f|\mathrm{j}_\nu^\dagger(\boldsymbol{k}')|i\rangle \, \delta(\omega + \omega_f - \omega_i) \tag{4}$$

and provides a definition of the relativistic MDFF $S_{\mu\nu}$ as the imaginary part of the 4-susceptibility $\chi_{\mu\nu}$ which can be expressed as a 4-current correlation function in the framework of the Kubo linear response theory. This expression constitutes the root of the relations bridging the world of condensed matter physics and optics. Finally, as guide for the reader, we summarized on table 1 the main quantities and equations derived in this paper as well as their relation between them.

For brevity, Hartree atomic units ($\hbar = e = m_e = 1$) and Lorentz-Heaviside units for the Maxwell equations will be used from now on, unless otherwise specified. The 3-vectors are labeled by roman letters and written in standard font as $\boldsymbol{x} \equiv x^a = (x^1, x^2, x^3)$. The 4-vectors are labeled by greek letters and written in roman font as $\mathrm{x}^\mu = (\mathrm{x}^0, \mathrm{x}^1, \mathrm{x}^2, \mathrm{x}^3) = (ct, \boldsymbol{x})$.

## 2 State of the art

Most of quantum relativistic scattering theories for TEM electrons have been developed for diffraction and holography [26, 34, 41, 42, 45, 46], i.e. elastic scattering, and only a few deal with electron energy loss spectroscopy [43, 44, 47], i.e. inelastic scattering. In this section, we will review the principal results and theories of the literature. For analyzing characteristic energy-losses of electrons in a TEM (or EELS) the detector is typically placed in the energy-dispersive plane of an energy filter in the far field of the sample, which allows to discriminate the scattered electrons according to their energy loss and scattering angle. Alternative modes excite the magnetic coils of the TEM such to image the achromatic plane of the filter or its reciprocal on the detector, which can be used to record an energy-filtered diffraction pattern or image. In the following we will focus on the conventional far-field setup, which allows probing the dispersion of characteristic excitations such as plasmons or core-losses, typically yielding the most interesting analysis of the solids electronic structure (e.g., including electron-energy-loss linear dichroism (ELD), electron-energy-loss magnetic chiral dichroism (EMCD) [51]).

The most widespread theory of EELS of optical excitations [12] is built upon a semiclassical description of the scattering process founded in a point-like description of the beam electrons. This notably implicates the initial and final electron states not being (stationary) energy eigenstates, which distinguishes this semiclassical approximation from quantum perturbation theory, where initial and final states are energy eigenstates. We recall that the most important result of the semiclassical formalism is that the retarded electron energy-loss probability $\Gamma^R$ appearing in the overall energy loss $\Delta E = \int_0^\infty \hbar\omega\Gamma^R(\omega)d\omega$ reads:

$$\Gamma^R(\omega) = \frac{4}{\hbar} \int d\mathbf{r}\,d\mathbf{r}'\text{Im}\left\{\mathbf{J}^*(\mathbf{r},\omega)\overleftrightarrow{\mathcal{G}}(\mathbf{r},\mathbf{r}',\omega)\mathbf{J}(\mathbf{r}',\omega)\right\}, \tag{5}$$

where $\overleftrightarrow{\mathcal{G}} = \overleftrightarrow{G} - \overleftrightarrow{G}_0$ is the screened electric Green dyadic of the classical Maxwell equations defined through [11, 52, 53]:

$$\boldsymbol{\nabla}\times\boldsymbol{\nabla}\times\overleftrightarrow{G}(\mathbf{r},\mathbf{r}',\omega) - k^2\epsilon(\mathbf{r},\omega)\overleftrightarrow{G}(\mathbf{r},\mathbf{r}',\omega) = -\frac{1}{c^2}\delta(\mathbf{r}-\mathbf{r}') \tag{6}$$

and the free Green's dyad:

$$\boldsymbol{\nabla}\times\boldsymbol{\nabla}\times\overleftrightarrow{G}_0(\mathbf{r},\mathbf{r}',\omega) - k^2\overleftrightarrow{G}_0(\mathbf{r},\mathbf{r}',\omega) = -\frac{1}{c^2}\delta(\mathbf{r}-\mathbf{r}'). \tag{7}$$

Here $\epsilon$ is the frequency dependent dielectric function, which is a local quantity in this classical setting (e.g., given by Drude-Lorentz theory). With the screened retarded Green's dyad the induced electric field is given by an external current source through

$$\mathbf{E}_{\text{ind}}(\mathbf{r},\omega) = -4\pi i\omega\int d\mathbf{r}'\,\overleftrightarrow{\mathcal{G}}(\mathbf{r},\mathbf{r}',\omega)\mathbf{J}_{\text{ext}}(\mathbf{r}',\omega). \tag{8}$$

Considering that the scattering predominantly occurs into very small angles around $z-$direction and that the energy loss in the low-loss regime is small compared to the total energy of the electron, we can write both the initial and final current as $\mathbf{J}(\mathbf{r},\omega) = \rho_0(\mathbf{r}_\perp,-q_z)e^{iq_z z}$. Here, $q_z = \omega/v$ and $\rho_0(\mathbf{r}_\perp,-q_z)$ denotes the Fourier transform of the density at $t = 0$ along $z-$direction and we assume that it is a real quantity (i.e. symmetric along the $z-$axis). With that the loss probability (5) can be further simplified to

$$\Gamma^R(\omega) = \frac{4}{\hbar}\int d\mathbf{r}_\perp\,d\mathbf{r}'_\perp\rho_0(\mathbf{r}_\perp,q_z)\rho_0(\mathbf{r}'_\perp,-q_z)\text{Im}\left\{\overleftrightarrow{\mathcal{G}}_{zz}(\mathbf{r}_\perp,\mathbf{r}'_\perp,q_z,-q_z,\omega)\right\}.$$

This equation exhibits the familiar structure of the first order perturbative approach depicted in Fig. 3. Indeed we can distinguish between the initial and final electron states and the (generalized) Green's function. This structure is frequently encountered throughout regardless of the actual derivation details (e.g., 1st order Born approximation, linear response, semiclassical approximation), since all of them employ the notation of a weak interaction at a certain stage. The quasistatic, non-retarded limit of the Green's dyad is obtained by solving the longitudinal part of the full Maxwellian response, Eq. (7), which corresponds to the solution of the first Maxwell equation.

In 1987, Echenique and collaborators proposed a quantum version of the quasistatic transition rate $\Gamma^{\text{QS}}_{i \to f}$ between initial and final beam electron state using a self-energy formalism [54]:

$$\Gamma^{\text{QS}}_{i \to f} = \frac{2}{\hbar} \sum_f \iint d\boldsymbol{r} \, d\boldsymbol{r}' \psi_f(\boldsymbol{r}) \psi_i^*(\boldsymbol{r}) \text{Im}\{-\mathcal{W}(\boldsymbol{r}, \boldsymbol{r}', \omega)\} \psi_f^*(\boldsymbol{r}') \psi_i(\boldsymbol{r}') \delta(\epsilon_f - \epsilon_i + \hbar\omega), \quad (9)$$

which corresponds to Fermi's golden rule as shown by García de Abajo [11]. Note that $\psi_i$ and $\psi_f$ are energy eigenstates of the electron probe with respective energies $\epsilon_i$ and $\epsilon_f$, which is one difference between the quantum and the classical treatment. $\mathcal{W}$ is the screened Green's function for the electrostatic potential defined as:

$$\phi_{\text{ind}}(\boldsymbol{r}', \omega) = \int d\boldsymbol{r} \, \mathcal{W}(\boldsymbol{r}', \boldsymbol{r}, \omega) \mathfrak{n}_{\text{ext}}(\boldsymbol{r}, \omega), \quad (10)$$

where $\phi_{\text{ind}}(\boldsymbol{r}', \omega)$ is the scalar potential induced at $\boldsymbol{r}'$ by a density of charges $\mathfrak{n}_{\text{ext}}(\boldsymbol{r}, \omega)$ located at $\boldsymbol{r}$. The imaginary part of it, $\text{Im}\{\mathcal{W}\}$, corresponds to the spectral density if the latter is a real quantity in the energy representation, which is typically the case (exceptions will be noted). The Green's function $\mathcal{W}$ or its imaginary part contain all the quantum mechanical information about excitations inside the target including valence as well as core excitations. It can thus be applied both for describing low-loss [55–57] and core-loss EELS.

Using linear response theory, García de Abajo proposed[1] an extension of Eq. (9) to the retarded regime [11]:

$$\Gamma^{\text{R}}_{i \to f} \sim -\sum_f \iint d\boldsymbol{r} \, d\boldsymbol{r}' \psi_f(\boldsymbol{r}) \boldsymbol{\nabla} \psi_i^*(\boldsymbol{r}) \text{Im}\{\overset{\leftrightarrow}{\mathcal{G}}(\boldsymbol{r}, \boldsymbol{r}', \omega)\} \psi_f^*(\boldsymbol{r}') \boldsymbol{\nabla}' \psi_i(\boldsymbol{r}') \delta(\epsilon_f - \epsilon_i + \hbar\omega). \quad (11)$$

This equation makes use of the paraxial approximation and has been used in several works [23, 58] in order to calculate the dichroism in the interaction between a vortex electron state and a (geometrically) chiral plasmonic nano-particle. We will show below how to generalize this equation making use of quantum propagators.

Although the above loss-probability and transition rate formulas are remarkably elegant and intuitive, they do not provide information on the propagation of the wavefunction in the microscope. However, a proper description of a phase-shaped EELS experiment requires the precise description of the illumination and detection systems. Moreover, information on the coherence of the electron beam, which plays a crucial role in holography, is not explicitly present in these expressions. They are therefore not sufficient to generally model EELS experiments in the TEM and represent certain limiting cases where the above effects may be neglected.

In order to describe these situations, another fundamental object has to be considered. This is the density matrix operator of the beam electrons, which in an energy eigenbasis $\{|\psi_n\rangle\}$ reads:

$$\hat{\rho} = \sum_n p_n |\psi_n\rangle \langle\psi_n|, \quad (12)$$

---

[1]To the best of our knowledge, there is no published demonstration of this formula.

where $p_n$ are the occupation probabilities associated to each state vector $n$. Inserting the completeness relation $\sum_r |r\rangle \langle r| = \mathbb{1}$, we obtain the fundamental tool for the description of wave optical experiments: the energy-dependent density matrix. It is defined as [15, 59] (see also Eq. (100)):

$$\rho(\boldsymbol{r}, \boldsymbol{r}', \omega) = \sum_n p_n \psi_n(\boldsymbol{r}) \psi_n^*(\boldsymbol{r}') \delta(\omega - \epsilon_n). \tag{13}$$

This quantity is particularly rich in terms of information; for example $I = \rho(\boldsymbol{r}, \boldsymbol{r})$ gives the intensity at coordinate $\boldsymbol{r}$ in position space (typically identified with (conjugated) image planes in the TEM). Even more importantly, the out-of-diagonal elements measure the mutual coherence of the electron field between positions $\boldsymbol{r}$ and $\boldsymbol{r}'$ [16]. In other words, non-zero off-diagonal terms entail electron interferences in the particular plane considered (which is defined by $z$ coordinate along the optical axis).

In 1993, Dudarev, Peng and Whelan [26] demonstrated (the different assumptions leading to this formula will be reviewed in details in this paper) that, in the quasi-static limit, the inelastic scattering of high energy electrons by a polarizable material can be described by the so-called kinetic equation:

$$\rho_f(\boldsymbol{r}, \boldsymbol{r}', E) = \mathscr{F}_{\boldsymbol{r}, -\boldsymbol{r}'} \left\{ \frac{S(\boldsymbol{k}, \boldsymbol{k}', \omega)}{k^2 k'^2} \right\} \rho_i(\boldsymbol{r}, \boldsymbol{r}', E + \hbar\omega), \tag{14}$$

where $\mathscr{F}$ denotes the Fourier transform, $\rho_i$ and $\rho_f$ are the density matrices of the electron probe before and after the interaction and $S(\boldsymbol{k}, \boldsymbol{k}', \omega)$ is the so-called *mixed dynamic form factor* (MDFF) [14] defined as :

$$S(\boldsymbol{k}, \boldsymbol{k}', \omega) = 2\pi \sum_f \langle i|\mathfrak{n}(\boldsymbol{k})|f\rangle \langle f|\mathfrak{n}^\dagger(\boldsymbol{k}')|i\rangle \, \delta(\epsilon_i - \epsilon_f + \hbar\omega), \tag{15}$$

where $\boldsymbol{k}$ is a wave-vector, $\mathfrak{n}$ is the electron density operator and $|f\rangle$ is an eigenbasis of the target electronic many-body state. Upon comparison with the quasistatic transition rate (9) we note that the MDFF is the Fourier space (and many-body) version of the spatial integral in said expression. It contains all the information on the correlation of the electronic charge density of the scatterer [16].

$S$ is a hermitian tensor, hence may be decomposed into a real-valued symmetric tensor containig information about the non-chiral (i.e., non-magnetic) transitions and an imaginary antisymmetric tensor (represesented by the vector S) containing the information about chiral (i.e., magnetic) transitions, reading [60]

$$S\left(\boldsymbol{k}, \boldsymbol{k}', \omega\right) = \boldsymbol{k} \cdot N(\omega) \cdot \boldsymbol{k}' + i(\boldsymbol{k} \times \boldsymbol{k}') \cdot \mathbf{S}(\omega) \tag{16}$$

in the dipole approximation (to be detailed further below). Note that the chiral transitions introduce an imaginary component here, which marks one of the few cases, where the spectral density is not a real object.

Remarkably, Eq. (14) shows that correlation are imprinted in the density matrix (and hence mutual coherence) of the beam during the scattering process. It leads to a fundamental principle of electron holography: generating interferences in order to trace back to the electronic correlations in the target. Indeed, EMCD experiments may be designed such to isolate the second term of Eq. (16), hence allowing to characterize magnetic materials in the TEM. It would be rather seducing to use such a formalism in the case of nano-optics and e.g. interpret EELS interference effects on surface plasmons in terms of electric field correlations measurements. The MDFF is particularly adapted to model core-loss spectroscopy as the measured phenomena appear to be quasi-static. However, it gives an incomplete picture of nano-optical

phenomena where retardation effects dramatically constrain the coherence properties of the field. Therefore, the definition of a relativistic MDFF and its link to nanooptical quantities is needed.

# 3 Preliminary remarks on the electromagnetic field

## 3.1 Gauge fixing and vacuum photon propagator

The Green's function of the Maxwell equation in the form given in the Annex (Eq. (191)) is the *vacuum photon propagator* $D^\mu_\nu$ defined by:

$$A^\mu(x') = \int dx\, D^\mu_\nu(x',x)J^\nu(x). \tag{17}$$

In the following, we are using Einstein summation convention, with greek letters for covariant indices and latin letters of spatial indices. For readability, all quantities and basic equations have been defined in the Annex. In order to calculate $D^\mu_\nu$ one needs to invert the Kernel in Eq. (191). Depending on the gauge, this task can require involved mathematical techniques as it may be singular. In nano-optics, principally three gauges for the electromagnetic field [61] are encountered in the literature: the Coulomb gauge, the (partial) Lorenz gauge and the temporal (or Weyl) gauge - each of them having different specific interests. We will therefore give $D^\mu_\nu$ in these cases [62, 63] only, but keeping in mind that the vacuum photon propagator can be expressed in arbitrary gauges:

- The Coulomb gauge corresponds to the condition:

$$\partial_i A^i = 0. \tag{18}$$

It is of particular interest in standard quantum electrodynamics as it enables a simple quantification of the potentials and leaving the Coulomb interaction in its classical and non-retarded form. In the Coulomb gauge the photon propagator reads [62, 63]:

$$
\begin{cases}
D^{ij} = \dfrac{4\pi}{k^2 - \frac{\omega^2}{c^2}}\left(\delta^{ij} + \dfrac{k^i k^j}{k^2}\right) & \text{(19a)} \\[3ex]
D^{00} = -\dfrac{4\pi}{k^2} & \text{(19b)} \\[2ex]
D^{i0} = 0 & \text{(19c)}
\end{cases}
$$

- The temporal gauge corresponds to the condition:

$$A^0 = 0. \tag{20}$$

It is particularly interesting because it drastically facilitates the calculation of conductivity in linear response theory. In the temporal gauge the photon propagator reads [62, 63]:

$$
\begin{cases}
D^{ij} = \dfrac{4\pi}{k^2 - \frac{\omega^2}{c^2}}\left(\delta^{ij} + \dfrac{k^i k^j}{\omega^2/c^2}\right) & \text{(21a)} \\[3ex]
D^{00} = 0 & \text{(21b)} \\[1ex]
D^{i0} = 0 & \text{(21c)}
\end{cases}
$$

- The Lorenz gauge corresponds to the condition:

$$\partial_\mu A^\mu = 0. \tag{22}$$

Its main interest is to decouple the motion equation for the four components of the potential. Indeed, in the Lorenz gauge, the propagator reads [63]:

$$D^{\mu\nu} = \frac{4\pi g^{\mu\nu}}{k^2 - \frac{\omega^2}{c^2}} \, . \tag{23}$$

## 3.2 Mutual coherence tensor, electromagnetic local and cross density of states

In Sec. 2, we introduced the Green dyadic $\overleftrightarrow{G}$ propagating a current source to an electric field as a function of the photon energy $\omega$. From this propagator, as it is commonly done in condensed matter [64], one can define a photonic spectral function (which in the case of optical fields is a tensor), usually called the *mutual coherence tensor* [49,50] (MCT), as:

$$\Lambda_{ij}(\boldsymbol{r},\boldsymbol{r}',\omega) = -\frac{2\omega}{\pi}\,\mathrm{Im}\left\{\overleftrightarrow{G}_{ij}(\boldsymbol{r},\boldsymbol{r}',\omega)\right\} \, . \tag{24}$$

From the fluctuation-dissipation theorem, the former quantity can be connected to the spectral correlations of the electromagnetic field [65,66]:

$$
\begin{aligned}
\mathcal{E}_i^j(\boldsymbol{r},\boldsymbol{r}',\omega) &= \frac{1}{2\pi}\int_{-\infty}^{+\infty} d\tau \; \langle E^i(\boldsymbol{r},t+\tau)E_j^*(\boldsymbol{r}',t)\rangle \, e^{-i\omega\tau} \\
&= \frac{8\omega^2 h}{c^2}\,\mathrm{Im}\left\{\overleftrightarrow{G}_{ij}(\boldsymbol{r},\boldsymbol{r}',\omega)\right\} \, .
\end{aligned}
\tag{25}
$$

The mutual coherence tensor is a pillar of nano-optics as it contains all the important information about the optical field. Indeed, from the early work of Agarwal [67], we know that by taking the trace of its diagonal elements (i.e. $\boldsymbol{r} = \boldsymbol{r}'$), we obtain the electromagnetic density of states (EMLDOS):

$$\mathcal{N}(\boldsymbol{r},\omega) = \sum_{i=1}^{3}\Lambda_{ii}(\boldsymbol{r},\boldsymbol{r},\omega), \tag{26}$$

where, for clarity, we made the summation explicitly appear. One can also consider partial electromagnetic density of states by only taking one of the components e.g. for $i \in \{1,2,3\}$:

$$\mathcal{N}_i(\boldsymbol{r},\omega) = \Lambda_{ii}(\boldsymbol{r},\boldsymbol{r},\omega). \tag{27}$$

The EMLDOS is involved in the description of a wide range of phenomena such as the Purcell effect [68] or the Casimir effect. In 2013, Cazé and collaborators introduced [69] the cross density of states (CDOS):

$$\mathcal{N}(\boldsymbol{r},\boldsymbol{r}',\omega) = \sum_{i=1}^{3}\Lambda_{ii}(\boldsymbol{r},\boldsymbol{r}',\omega), \tag{28}$$

which of course allows the definition of a partial CDOS e.g. for $i \in \{1,2,3\}$:

$$\mathcal{N}_i(\boldsymbol{r},\boldsymbol{r}',\omega) = \Lambda_{ii}(\boldsymbol{r},\boldsymbol{r}',\omega). \tag{29}$$

From Eq. (25), it is clear that the CDOS measures the electromagnetic correlations between two points in space at the energy $\omega$. As extensively discussed in [50,69,70], this quantity is particularly relevant to characterize the spatial coherence of optical field and study e.g. the Anderson localization in random media.

It is also quite standard to normalize the CDOS by the EMLDOS, thus defining a new quantity:

$$\mathcal{C}(\boldsymbol{r},\boldsymbol{r}',\omega) = \frac{\mathcal{N}(\boldsymbol{r},\boldsymbol{r}',\omega)}{\sqrt{\mathcal{N}(\boldsymbol{r},\omega)\mathcal{N}(\boldsymbol{r}',\omega)}} \, , \tag{30}$$

which, depending on the context and for historical reasons, is referred to as complex degree of coherence [49] or mode connectivity [70]. Thanks to the Schwarz inequality, one can show that $0 \leq \mathcal{C} \leq 1$ where the case $\mathcal{C} = 0$ corresponds to uncorrelated points and $\mathcal{C} = 1$ to the situation of strong connection [70]. Extension to polarization-dependent correlations is straightforward from $\Lambda_{ij}$ and leads e.g. to the definition of a complex degree of mutual polarization [71].

No matter the plethora of quantities defined in the literature over the last decades, it is crucial to state that they are all contained in the most general mutual coherence tensor $\Lambda_{ij}$.

# 4 Fundamentals of Scalar Relativistic Quantum Electrodynamics

In this section we shortly recapitulate the fundamentals of relativistic QED in order to obtain very general covariant (i.e., fully retarded) expression for the inelastic transition matrix elements and transition rates, which are then further detailed in the following chapter. For the sake of clarity we explicitly note the elementary charge $e$ to highlight the perturbation order of the theory. Moreover, we employ upper and lower indices to indicated covariant and contravariant vectors in this section.

As stated previously we can largely neglect the ramifications of the electron spin in the inelastic scattering process as the spin-orbit coupling is small at the large electron velocities considered here. A scalar relativistic formulation of the scattering process is therefore largely sufficient for our purpose. The fundamentals of a scalar relativistic QED, i.e., the Feynman rules following from a perturbative treatment of the Klein-Gordon field weakly coupled to the electromagnetic field,

$$\left(\partial^\mu \partial_\mu + m_e^2\right)\psi = -ie\left(\partial^\mu A_\mu + A^\mu \partial_\mu\right)\psi - e^2 A^\mu A_\mu, \tag{31}$$

are derived, e.g., in the book of Greiner [72].

Here, we only repeat the fundamentals of scalar relativistic perturbation theory important for the following computations. First, we restrict ourselves to the first order of the perturbation (i.e., scattering matrix terms linear in the interaction constant $e^2$) as higher order terms such as Vertex corrections can be expected to be very small (their relative strength corresponds to those of the related Lamb shift). This allows us to neglect the quadratic coupling term on the right hand side of Eq. (31). The same reasoning allows us to get rid of the quadratic term occurring in the definition of the transition current:

$$\begin{aligned} J_\mu &= ie\psi_f^* \overleftrightarrow{\partial}_\mu \psi_i - 2e^2 A_\mu \psi_f^* \psi_i \\ &\approx ie\left(\psi_f^* \partial_\mu \psi_i - \psi_i \partial_\mu \psi_f^*\right), \end{aligned} \tag{32}$$

which appears in the calculation of the vacuum photon propagator $D$ (Eq. (17)). We finish our preparations with noting the scalar product as employed within the framework of scalar relativistic QED

$$\left(\psi_f | \psi_i\right) = \int d^3 r \psi_f^*(r) i \overleftrightarrow{\partial_0} \psi_i(r) \tag{33}$$

and introducing the scattering matrix

$$\mathcal{S}_{fi} = \lim_{t \to +\infty} \langle \psi_f | U(t, -\infty) | \psi_i \rangle, \tag{34}$$

which describes the transition of an initial beam electron state $\psi_i$ to a final state $\psi_f$ under an influence of the time evolution operator $U(t_2, t_1)$ containing the actual physics of scattering process. In other words we have

$$|\psi_f\rangle = \mathcal{S}_{fi} |\psi_i\rangle \tag{35}$$

and

$$\hat{\rho}_f = \mathcal{S}_{fi}\mathcal{S}_{fi}^*\hat{\rho}_i \tag{36}$$

for the transition of a state vector (i.e. wave function) and the density operator (i.e., density matrix). The latter is the abstract retarded version of the kinetic Eq. (14) noted previously. With these tools and the corresponding Feynman rules, we may now note the scattering matrix for the first order scattering process depicted in Fig. 4(a)

$$\begin{aligned}
\mathcal{S}_{fi} &= -e\int dr\psi_f^*\left(\overleftarrow{\partial}^\mu A_\mu - A_\mu\overrightarrow{\partial}^\mu\right)\psi_i \\
&= -ie\int d^4r J_{fi}^\mu(r)A_\mu(r)\,.
\end{aligned} \tag{37}$$

The first line contains the derivatives of the beam electron wave function reminiscent of a symmetric version of Garcia de Abajo's formula Eq. (11). On the second line, the relation to the transition current is highlighted, which provides an *ab-initio* ratio for to the current-current coupling terms in the linear-response regime (Kubo formalism) discussed further below. Indeed, the transition probability per electron into some final state reads

$$\Gamma_{i\to f} = \left|\mathcal{S}_{fi}\right|^2\,, \tag{38}$$

which transforms into

$$\Gamma_{i\to f} = -e^2\int dr\,dr'J_{fi}^\mu(r)\underbrace{A_\mu(r)A_\nu^*(r')}_{\text{Im}\{\mathcal{D}_{\mu\nu}(r,r')\}}J_{fi}^{*\nu}(r')\,, \tag{39}$$

upon inserting Eq. (37). Here we introduced a new abbreviation for the correlation of two vector potentials. This quantity represents the central part (i.e., between the two outer vertices) of Fig. 4(b) and hence may be directly related to the four-susceptibility (i.e., the central loop the Feynman graph in Fig. 4(b)) :

$$\chi_{fi}^{\kappa\lambda}(r,r') = -ie^2\theta(r_0-r_0')\underbrace{\xi_i^*(r)\overleftrightarrow{\partial^\kappa}\xi_f(r)}_{j^\kappa(r)}\underbrace{\xi_f^*(r')\overleftrightarrow{\partial^\lambda}\xi_i(r')}_{j^{*\lambda}(r')} \tag{40}$$

via

$$\mathcal{D}_{\mu\nu}(r,r') = \int du\,du'\,D_{\mu\kappa}(r-u)D_{\nu\lambda}^*(r'-u')\chi^{\kappa\lambda}(u,u')\,, \tag{41}$$

where $\xi_i,\xi_f$ respectively denote the initial and final states in the target. In the last expression we summed over all final states, i.e., $\chi = \sum_f\chi_{fi}$, to take into account every possible final scattering state of the target while the initial state is taken as the fundamental state. The latter expression may be transformed into reciprocal space taking into account the homogeneity of the interaction in time, yielding

$$\mathcal{D}_{\mu\nu}(\boldsymbol{k},\boldsymbol{k}',\omega) = \frac{4\pi g_{\mu\kappa}}{\boldsymbol{k}^2-\omega^2}\frac{4\pi g_{\nu\lambda}}{\boldsymbol{k}'^2-\omega^2}\chi^{\kappa\lambda}(\boldsymbol{k},\boldsymbol{k}',\omega) \tag{42}$$

if employing the photon propagator in Lorentz gauge (Eq. (23)). Indeed, there is a one-to-one relationship between the spectral representation of $\chi$ and $\mathcal{D}$ and the relativistic generalization of the mixed dynamic form factor

$$S_{\mu\nu}(\boldsymbol{k},\boldsymbol{k}',\omega) = 2\pi\sum_f\langle i|j_\mu(\boldsymbol{k})|f\rangle\langle f|j_\nu^\dagger(\boldsymbol{k}')|i\rangle\,\delta(\omega+\omega_f-\omega_i), \tag{43}$$

where we introduced the 4-current operator $j_\mu(\boldsymbol{k})$ in reciprocal space. The relationship can be demonstrated by employing the Dirac identity $\dfrac{1}{\omega \pm i0^+} = \mp i\pi\delta(\omega) + \mathcal{P}\dfrac{1}{\omega}$ yielding [73]

$$\text{Im}\{-\chi_{\mu\nu}(\boldsymbol{k},\boldsymbol{k}',\omega)\} = S_{\mu\nu}(\boldsymbol{k},\boldsymbol{k}',\omega), \tag{44}$$

and hence

$$\text{Im}\{\mathcal{D}_{\mu\nu}(\boldsymbol{r},\boldsymbol{r}',\omega)\} = \mathscr{F}^{-1}_{\boldsymbol{r},-\boldsymbol{r}'} \left\{ \frac{\mathcal{S}_{\mu\nu}(\boldsymbol{k},\boldsymbol{k}',\omega)}{(k^2-\omega^2)(k'^2-\omega^2)} \right\}. \tag{45}$$

Note that we again made the assumption that the MDFF is a real quantity here. Indeed, this assumption allows us to to drop the difference between causal Green's functions appearing in the diagrammatic perturbation theory (i.e., Feynman diagrams used in this section) and retarded Green's function in linear response formalism used further below. In the most general case this difference must be treated carefully, leading, e.g., to a more complicated relationship between retarded (and advanced) 4-susceptibility and the MDFF

$$\frac{i}{2}\left(\chi^{\text{R}}_{\mu\nu}(\omega+i0^+) - \chi^{\text{A}}_{\mu\nu}(\omega+i0^-)\right) = S_{\mu\nu}(\omega). \tag{46}$$

The transition probability for a particular energy loss finally reads:

$$\Gamma_{i\to f} = 2\pi \int d\boldsymbol{r}\,d\boldsymbol{r}'\,J^\mu_{fi}(\boldsymbol{r})J^{\nu*}_{fi}(\boldsymbol{r}')\,\mathscr{F}^{-1}_{\boldsymbol{r},-\boldsymbol{r}'}\left\{ \frac{\mathcal{S}_{\mu\nu}(\boldsymbol{k},\boldsymbol{k}',\omega)}{(k^2-\omega^2)(k'^2-\omega^2)} \right\}. \tag{47}$$

In Sec. 5 we will pick up these strands and further transform this expression to obtain a generalized retarded expression of Eq. (11) noted previously. Additionally, we will show how to derive the celebrated magic angle correction [74, 75, 75–77] in Appendix C.

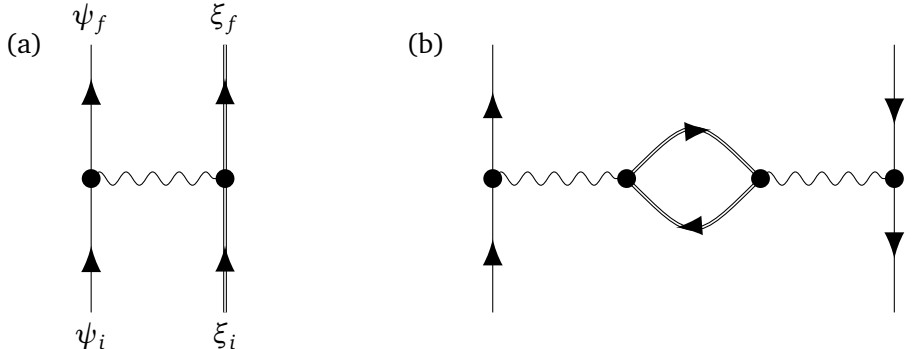

Figure 4: Diagram representation of the first-order inelastic electron-target scattering: (a) an electron of wavefunction $\psi(r,t)$ interacts with a target represented by a target many-body wave function $\xi(r,t)$ by exchanging free virtual photons. (b) the beam electron density matrix is modified through a modified photon propagator containing the target correlation function.

# 5 Propagators for the electromagnetic field in presence of a polarizable medium

In this section, we focus on the propagator for the EM field in the presence of a polarizable medium, e.g. a metallic nano-particle. For completeness, we first consider the quasi-static

case and show that the screened interaction can be connected to the MDFF, see Sec. 5.1.1. This way, we connect the standard formalism of EELS to optical quantities.

In Sec. 5.2, using a Dyson development, we calculate the exact photon propagator in the presence of a polarizable material. In a complete analogy to what has been done with the MDFF, we then connect this photonic kernel to the charge and current density correlation functions of the scatterer.

## 5.1 Quasistatic approach: modification of the Coulomb propagator, electron density correlation function

We first consider the quasi-static limit $c \to \infty$. In this situation, the calculation of the EM field propagation simply reduces to the resolution of the Poisson equation. Thus, as illustrated on Fig. 5, we simply need to consider the scalar potential and the electron charge density of the target.

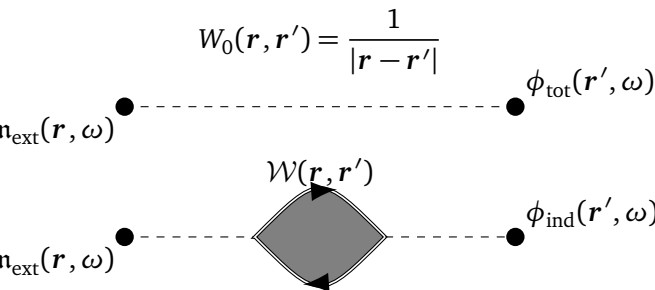

Figure 5: Schematics illustrating the problem tackled in this section. The Green function for the electrostatic potential in vacuum is simply the Coulomb propagator. For our purpose (i.e., to compute the screened interaction), this Green function needs to be modified.

In vacuum the potential $\phi_{ind}$ induced in $\boldsymbol{r}'$ by an external charge $\rho_{ext}$ in $\boldsymbol{r}$ is simply given by the Coulomb law. In other terms, the free-space EM propagator is simply given by $W_0(\boldsymbol{r}, \boldsymbol{r}', \omega) = 1/|\boldsymbol{r} - \boldsymbol{r}'|$. This law needs to be modified in the presence of a dielectric medium in order to take into account e.g. the screening effect in the material. Particularly we expect the new propagator $\mathcal{W}$ to be energy-dependent as, contrary to the vacuum, the target can be dispersive.

In this section, we will derive the new propagator $\mathcal{W}$ and connect it to the mixed dynamic form factor.

### 5.1.1 Linear response electrostatic susceptibility

In the following, we define the electronic charge density operator $\mathfrak{n}$ for the target as $\mathfrak{n}(\boldsymbol{r}) = e\hat{n}(\boldsymbol{r})$ where $\hat{n}$ is the particle number operator for the electrons. We first need to calculate the response of the target of electronic density $\mathfrak{n}(\boldsymbol{r}, t)$ to an external perturbation $\phi_{ext}(\boldsymbol{r}, t)$. The electronic charge $\delta\langle \mathfrak{n}(\boldsymbol{r}, t)\rangle \equiv \mathfrak{n}_{ind}(\boldsymbol{r}, t) = \langle \mathfrak{n}(\boldsymbol{r}, t)\rangle - \langle \mathfrak{n}(\boldsymbol{r}, t)\rangle_0$ induced on the target by this electrostatic field can be calculated using the Kubo formula [78,79]:

$$\mathfrak{n}_{ind}(\boldsymbol{r}, t) = -i \int_{t_0}^{\infty} dt' \, \theta(t - t') \langle [\mathfrak{n}(\boldsymbol{r}, t), H(t')] \rangle_0 \, e^{-\eta(t - t')}, \qquad (48)$$

where $\eta \to 0^+$, $t_0$ the starting time of the interaction and $H(t)$ is the perturbation Hamiltonian given by:

$$H(t) = \int_{\mathbb{R}^3} d\boldsymbol{r}' \, \mathfrak{n}(\boldsymbol{r}',t)\phi_{\text{ext}}(\boldsymbol{r}',t). \tag{49}$$

Therefore, one can write:

$$\mathfrak{n}_{\text{ind}}(\boldsymbol{r},t) = \int_0^\infty dt' \int_{\mathbb{R}^3} d\boldsymbol{r}' \left\{ -\frac{i}{\hbar}\theta(t-t')\langle[\mathfrak{n}(\boldsymbol{r},t),\mathfrak{n}(\boldsymbol{r}',t')]\rangle_0 \right\} \phi_{\text{ext}}(\boldsymbol{r}',t')e^{-\eta(t-t')}. \tag{50}$$

Besides, the linear-response electric susceptibility $\chi$ is implicitly defined as:

$$\mathfrak{n}_{\text{ind}}(\boldsymbol{r},t) = \int d\boldsymbol{r}' \int dt' \, \chi(\boldsymbol{r},\boldsymbol{r}',t,t')\phi_{\text{ext}}(\boldsymbol{r}',t'). \tag{51}$$

Comparing equations (50) and (51), one can deduce the following expression for $\chi$:

$$\chi(\boldsymbol{r},\boldsymbol{r}',t,t') = -i\theta(t-t')\langle[\mathfrak{n}(\boldsymbol{r},t),\mathfrak{n}(\boldsymbol{r}',t')]\rangle_0. \tag{52}$$

We retrieve the well-known linear response susceptibility at equilibrium in the real space. In the spectral domain, the electrostatic susceptibility reads:

$$\text{Im}\{\chi(\boldsymbol{r},\boldsymbol{r}',\omega)\} = -\frac{\pi}{Z}\sum_{n,n'}\langle n|\mathfrak{n}(\boldsymbol{r})|n'\rangle\langle n'|\mathfrak{n}(\boldsymbol{r}')|n\rangle e^{-\beta\hbar\omega_n}$$
$$\times\left(1+e^{-\beta\hbar\omega}\right)\delta(\omega+\omega_n-\omega_{n'}). \tag{53}$$

The latter equation is valid for any temperature $T = 1/(\beta k_B)$, with $k_B$ the Boltzmann constant, as soon as we are at thermal equilibrium. We now take the limit of the latter expression in the limit of null temperature $T = 0$. This is fully justified when the energy of the electronic excitations are significantly greater than the thermal energy at room temperature $k_{\text{B}}T \approx 25$ meV. This will be the case in the following developments because e.g. the energy of SPs is typical energy of 1 eV. In equation (53), we can then replace $\frac{1}{Z}\sum_n \langle n|.|n\rangle \exp(-\beta\hbar\omega_n) \to \langle 0|.|0\rangle$ and $\beta \to 0$ which gives:

$$\text{Im}\{-\chi(\boldsymbol{r},\boldsymbol{r}',\omega)\} = 2\pi\sum_n \langle 0|\mathfrak{n}(\boldsymbol{r})|n\rangle\langle n|\mathfrak{n}(\boldsymbol{r}')|0\rangle\delta(\omega+\omega_n-\omega_0). \tag{54}$$

Therefore we see that the latter corresponds to the Fourier transform of the MDFF (15).

### 5.1.2 Connection between the mixed dynamic form factor and the screened interaction

Now, we are in position to calculate the screened electrostatic propagator $W$ which is formally defined such that

$$\mathcal{W}(\boldsymbol{r},\boldsymbol{r}',\omega) = \int d\boldsymbol{r}_1 \int d\boldsymbol{r}_2 \, \frac{\chi(\boldsymbol{r}_1,\boldsymbol{r}_2,\omega)}{|\boldsymbol{r}-\boldsymbol{r}_1||\boldsymbol{r}'-\boldsymbol{r}_2|}, \tag{55}$$

where we used the time-translation invariance of $W_0$ and $\chi$ to simply express the Fourier transform. The latter can be Fourier transformed with respect to $\boldsymbol{r}$ and $\boldsymbol{r}'$ which leads to:

$$\mathcal{W}(\boldsymbol{k},-\boldsymbol{k}',\omega) = (4\pi)^2 \frac{\chi(\boldsymbol{k},-\boldsymbol{k}',\omega)}{k^2 k'^2}. \tag{56}$$

We used the identity [80] $\mathscr{F}_k\left\{\frac{1}{r}\right\} = \frac{4\pi}{k^2}$ where $\mathscr{F}$ denotes the Fourier transform. Nevertheless, the screened potential and the electric Green dyadic are related in the quasistatic regime by [81]:

$$\overleftrightarrow{\mathcal{G}}(\boldsymbol{r},\boldsymbol{r}',\omega) = \frac{1}{4\pi\omega^2}\boldsymbol{\nabla}\boldsymbol{\nabla}'\mathcal{W}(\boldsymbol{r},\boldsymbol{r}',\omega). \tag{57}$$

Moreover, Fourier transforming the latter gives:

$$\overleftrightarrow{\mathcal{G}}(\boldsymbol{k}, -\boldsymbol{k}', \omega) = \frac{1}{4\pi\omega^2}\boldsymbol{k}\boldsymbol{k}'\mathcal{W}(\boldsymbol{k}, -\boldsymbol{k}', \omega). \tag{58}$$

Therefore, using Eq. (56) we get:

$$\overleftrightarrow{\mathcal{G}}(\boldsymbol{k}, -\boldsymbol{k}', \omega) = \frac{4\pi}{\omega^2}\frac{\boldsymbol{k}}{k^2}\frac{-\boldsymbol{k}'}{k'^2}\,\chi(\boldsymbol{k}, -\boldsymbol{k}', \omega). \tag{59}$$

We also need to calculate the imaginary part of the screened interaction as it is involved in the definition of the loss probability (9). Therefore, taking the imaginary part of (55) and by using Eq. (54), we get:

$$\mathrm{Im}\{-\mathcal{W}(\boldsymbol{r}, \boldsymbol{r}', \omega)\} = 2\pi\sum_n \int d\boldsymbol{r}_1 \int d\boldsymbol{r}_2 \frac{\langle 0|\rho(\boldsymbol{r}_1)|n\rangle\langle n|\rho(\boldsymbol{r}_2)|0\rangle}{|\boldsymbol{r}-\boldsymbol{r}_1||\boldsymbol{r}'-\boldsymbol{r}_2|}\delta(\hbar\omega+\hbar\omega_n-\hbar\omega_0). \tag{60}$$

Finally using the latter equation combined together with Eq. (55) and (15), we get:

$$\mathrm{Im}\{-\mathcal{W}(\boldsymbol{r}, \boldsymbol{r}', \omega)\} = \frac{2}{\pi}\mathscr{F}_{\boldsymbol{k},-\boldsymbol{k}'}\left[\frac{S(\boldsymbol{k}, -\boldsymbol{k}', \omega)}{k^2 k'^2}\right]. \tag{61}$$

A quick look to the latter formulate clearly indicates that, as expected, the kernel of Eq. (9) and (14) are the same. Depending on the situation investigated, each of these kernels can be interchangeably used:
• When ab initio calculations are required (typically in the case of core-loss spectroscopy), one will preferably use the MDFF as it explicitly displays the quantum mechanical charge density correlations.
• When classical photonic systems are investigated, one will preferably use the screened interaction as it can be simply calculated by e.g. boundary element method [82, 83].

Finally, by using the definition of the mutual coherence tensor (24) together with Eq. (59), we obtain:

$$\overleftrightarrow{\Lambda}(\boldsymbol{r}, \boldsymbol{r}', \omega) = \frac{4\omega}{\pi^2}\mathscr{F}_{\boldsymbol{k},-\boldsymbol{k}'}\left[\frac{\boldsymbol{k}\boldsymbol{k}'\,S(\boldsymbol{k}, -\boldsymbol{k}', \omega)}{k^2 k'^2}\right]. \tag{62}$$

This equation shows that, in the quasi-static limit, the CDOS and the EMLDOS are respectively given by the MDFF and the dynamic form factor (DFF, [14]). In other words, in this limit, the electric field correlations (encoded in the MCT) simply reproduce the electronic charge correlations in the target (encoded in the MDFF). This result is of course expected as, in the quasi-static limit, the electric field and the charge density are simply related through:

$$\boldsymbol{E}(\boldsymbol{r}, t) = -\boldsymbol{\nabla}\int d\boldsymbol{r}'dt\, W(\boldsymbol{r}, \boldsymbol{r}', t, t')\rho(\boldsymbol{r}', t'). \tag{63}$$

Although intuitive, Eq. (61) and (62) have, to the best of our knowledge, never been derived. It enables to put on the same level the MDFF formalism for the electronic correlations [15, 16, 19] and the MCT formalism for the photonic correlations [65, 69, 84].

## 5.2 Retarded approach: Photon propagator and electron four-current correlation function

We now turn to the retarded case where both the scalar $\phi$ and the vector potential $\boldsymbol{A}$ need to be considered. In his wonderful review [11], García de Abajo suggested to use the Kubo formalism for the current density to derive a retarded form of the latter equations. However, we

could not find such a demonstration in the literature; therefore, in this section, we will follow this suggestion and derive a retarded version of the linear response formalism established in the last section.

The main difficulty in the retarded regime is the choice of the gauge. The developments found in the literature use different choices of gauge depending on the problem, so that a straightforward application is not possible. Some gauge choices are particularly convenient to calculate the EM field in vacuum e.g. the Coulomb gauge. However, these choices may, on the other hand, harden the calculation of the response function of the material. In order to avoid this difficulty while keeping a compact formalism, we will carry, when necessary, the calculation with four-vectors.

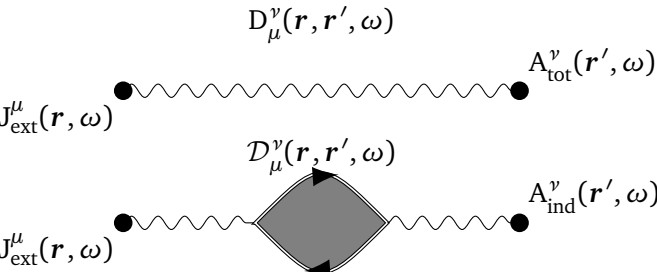

Figure 6: Schematics illustrating the problem tackled in this section. The Green function for the electromagnetic field in vacuum is simply the photon propagator which expression depends on the gauge choice. For our purposes (i.e., to compute the screened interaction), this Green function needs to be modified.

In vacuum, the 4-current $J_{ext}^{\nu}$ at x will generate a potential $A_{ind}^{\mu}$ at x′ which is related by the vacuum photon propagator $D_{\nu}^{\mu}(x',x)$, see Eq. (17). In the same way as in the quasi-static regime, the presence of a polarizable material will modify the EM propagator. The goal of this section is therefore to calculate the exact photon propagator $\mathcal{D}_{\mu}^{\nu}$ in the presence of the target as illustrated on Fig. 6. However, contrary to the quasi-static case, we need to take into account both the induced *charge and current* densities in the medium which are compactly represented by the 4-current density $J^{\mu}$.

### 5.2.1 Linear response electromagnetic susceptibility

The main interest of the previous quasi-static developments is that, now, the retarded case can be straightforwardly treated by analogy. Particularly, the 4-current $\delta\langle J^{\nu}(\boldsymbol{r},t)\rangle \equiv J_{ind}^{\nu}(\boldsymbol{r},t) = \langle J^{\nu}(\boldsymbol{r},t)\rangle - \langle J^{\nu}(\boldsymbol{r},t)\rangle_0$ induced in the medium by an external perturbation $A_{ext}^{\nu}$ is given by the Kubo formula:

$$J_{ind}^{\nu}(\boldsymbol{r},t) = -i \int_{t_0}^{\infty} dt' \, \theta(t-t') \langle [J^{\nu}(\boldsymbol{r},t), H(t')]\rangle_0 \, e^{-\eta(t-t')}, \tag{64}$$

where $\eta$ is an infinitely small positive real number. The perturbation Hamiltonian is then given by:

$$H(t') = \int d\boldsymbol{r}' J^{\mu}(\boldsymbol{r}',t') A_{\mu}(\boldsymbol{r}',t'). \tag{65}$$

Substituting the latter in the former, we get:

$$J_{\text{ind}}^{\nu}(\boldsymbol{r},t) = -i\theta(t-t')\int dt'\int d\boldsymbol{r}'\big\langle\big[J^{\nu}(\boldsymbol{r},t),J_{\mu}(\boldsymbol{r}',t')\big]\big\rangle_0 A_{\mu}^{\text{ext}}(\boldsymbol{r}',t'), \qquad (66)$$

we can then define a four-susceptibility $\chi_{\mu}^{\nu}$ as:

$$J_{\text{ind}}^{\nu}(\boldsymbol{r},t) = \int dt'\int d\boldsymbol{r}'\,\chi_{\mu}^{\nu}(\boldsymbol{r},\boldsymbol{r}',t,t')A_{\mu}^{\text{ext}}(\boldsymbol{r}',t') \qquad (67)$$

from which we deduce the linear response four-susceptibility tensor:

$$\chi_{\mu}^{\nu}(\boldsymbol{r},\boldsymbol{r}',t,t') = -i\theta(t-t')\big\langle\big[J^{\nu}(\boldsymbol{r},t),J_{\mu}(\boldsymbol{r}',t')\big]\big\rangle_0. \qquad (68)$$

Note that gauge invariance and four-current conservation imply $\partial^{\mu}\chi_{\mu}^{\nu} = \partial_{\nu}\chi_{\mu}^{\nu} = 0$ [73], a property, which we will dwell upon in the following. The structure of (68) being exactly analogue to (52), we can immediately deduce the spectral representation of the four-susceptibility at $T = 0$:

$$\chi_{\mu}^{\nu}(\boldsymbol{r},\boldsymbol{r}',\omega) = 2\sum_n \frac{\langle 0|J^{\nu}(\boldsymbol{r})|n\rangle\,\langle n|J_{\mu}(\boldsymbol{r}')|0\rangle}{\omega + \omega_n - \omega_0 + i\eta}. \qquad (69)$$

From Eq. (68), one can see that the linear-response four susceptibility has the following structure:

$$\chi_{\mu}^{\nu} = \begin{pmatrix} C_{\rho,\rho} & C_{\rho,j^b} \\[2mm] \hline \\[-2mm] C_{j_a,\rho} & C_{j_a,j^b} \end{pmatrix}, \qquad (70)$$

where we recall that $C_{\hat{A},\hat{B}}$ denotes the correlator between two quantities $\hat{A}$ and $\hat{B}$ being either/or $\rho$ and $j$. The diagonal elements of this tensor are therefore the charge-charge and current-current correlators while the out of diagonal elements correspond to charge-current correlators.

Let's stress an important semantic point. Both susceptibilities (52) and (68) are called retarded as they involved retarded electronic Green functions defined as (189). Nevertheless, let's keep in mind that, in our case, the retardation needs to be understood in the sense of the EM field, the regime is therefore defined by the value taken for $c$. To summarize:

• In the quasi-static regime ($c \to \infty$), the problem reduces to the Poisson equation and only the scalar potential and charge densities play a role in the response of the system. The light-matter interaction Hamiltonian is then taken to be (49) and the response function of the target is determined, to the first order, by charge-charge correlations.

• In the retarded regime ($c$ finite), both scalar and vector potentials need to be considered and the light-matter interaction Hamiltonian is then (65). The problem essentially reduces to a choice of gauge. If one is interested in e.g. the conduction properties of a metal, a suitable choice would be to use the temporal gauge $\phi = 0$ where the electric field is fully determined by the vector potential $\boldsymbol{E} = -i(\omega/c)\boldsymbol{A}$. In this case, the conductivity tensor defined as:

$$E^a(\boldsymbol{r},t) = \iint d\boldsymbol{r}'dt'\,\sigma_b^a(\boldsymbol{r},\boldsymbol{r}',t,t')j^b(\boldsymbol{r}',t') \qquad (71)$$

can be straightforwardly obtained by the Kubo formula and gives:

$$\text{Re}\big\{-\sigma_b^a(\boldsymbol{r},\boldsymbol{r}',\omega)\big\} = \frac{2\pi c}{\omega}\sum_n \langle 0|j^a(\boldsymbol{r})|n\rangle\,\langle n|j_b(\boldsymbol{r}')|0\rangle\,\delta(\omega+\omega_n-\omega_0). \qquad (72)$$

If one chooses a gauge where both $A$ and $\phi$ are non-zero, both the temporal and spatial parts of the Hamiltonian need to be considered and conductivity would include charge densities in its definition.

However, although the temporal gauge seems to simplify the situation on the electronic level, it complicates the expression of the photon propagators. In fact, in our case, where both electrons and photon propagation needs to be taken into account, no gauge seems to give a dramatically simpler solution.

### 5.2.2 Retarded electric Green dyadic

We are now in position to calculate the propagator for the EM field in presence of the polarizable medium. Thus, we consider the situation described in the introduction of this section: an external source term represented by the four-current $J^\mu_{\text{ext}}$ is positioned at $r$ and we want to calculate the total four-potential $A^\nu_{\text{tot}}$ induced at $r'$. Like we did in the quasi-static case, we apply the Born approximation (see diagram Fig. 6) and get the *retarded screened interaction*:

$$\mathcal{D}^\beta_\alpha(b,a) = D^\beta_\mu(b,\overline{2})\,\chi^\mu_\nu(\overline{2},\overline{1})\,D^\nu_\alpha(\overline{1},a),\tag{73}$$

keeping in mind that there is an implicit summation on the repeated indexes.
In nano-optics, we commonly work with electric and magnetic fields so that the electric Green dyadic $\overset{\leftrightarrow}{G}$ is one of the most important and fundamental object of the theory. These objects (fields and dyadic) have the advantage of being gauge-independent. However, so far we worked with the potentials $(\phi, A)$ as they have simpler transformation laws and symmetries; moreover, they strongly facilitate the connection with the many-particle Kubo formalism. Nevertheless, in this section, we will derive the electric Green dyadic using the results of the previous section in order to obtain formula adapted to discuss nano-optical experiments.

Combining the definition of the Faraday tensor (193) and the one of the photon propagator (17), we can write:

$$F^{\epsilon\gamma}(x') = \int d^4x \left[ \partial'^\epsilon \mathcal{D}^\gamma_\alpha(x',x) - \partial'^\gamma \mathcal{D}^\epsilon_\alpha(x',x) \right] J^\alpha(x),\tag{74}$$

where the prime in $\partial'^\epsilon$ indicates that the derivative is taken with respect to $x'$. Besides, from the explicit form of the Faraday tensor (194), one can directly deduce that the component $E^i$ of the electric field is given by:

$$E^i = F^{i0} = -F^{0i} = \partial^i A^0 + \partial^0 A^i.\tag{75}$$

Therefore, using (74) we get:

$$E^i(x') = \int d^4x \left( \partial'^i \mathcal{D}^0_\alpha(x',x) - \partial'^0 \mathcal{D}^i_\alpha(x',x) \right) J^\alpha(x).\tag{76}$$

Now, we decompose the sums over $\alpha$ as:

$$\mathcal{D}^i_\alpha(x',x) J^\alpha(x) = \mathcal{D}^i_0(x',x) J^0(x) - \mathcal{D}^i_a(x',x) J^a(x)\tag{77}$$

and,

$$\mathcal{D}^0_\alpha(x',x) J^\alpha(x) = \mathcal{D}^0_0(x',x) J^0(x) - \mathcal{D}^0_a(x',x) J^a(x).\tag{78}$$

Besides, one can write the continuity equation as:

$$\partial_\mu J^\mu = \left(\tfrac{\partial_t}{c}, -\boldsymbol{\nabla}\right).(c\rho, \boldsymbol{j}) = \partial_t \rho + \boldsymbol{\nabla}.\boldsymbol{j} = 0.\tag{79}$$

Fourier transforming in time the latter gives:

$$-i\omega\rho + ik_a J^a = 0\,, \tag{80}$$

which finally gives:

$$J^0 = c\rho = \frac{ck_a}{\omega}J^a\,. \tag{81}$$

Fourier transforming in space equations (77) and (78) and using the latter equation, we get:

$$\mathcal{D}_\alpha^i(\mathbf{k}',\mathbf{k})J^\alpha(\mathbf{k}) = \frac{c}{\omega}\mathcal{D}_0^i(\mathbf{k}',\mathbf{k})k_jJ^j(\mathbf{k}) - \mathcal{D}_a^i(\mathbf{k}',\mathbf{k})J^a(\mathbf{k}) \tag{82}$$

and,

$$\mathcal{D}_\alpha^0(\mathbf{k}',\mathbf{k})J^\alpha(\mathbf{k}) = \frac{c}{\omega}\mathcal{D}_0^0(\mathbf{k}',\mathbf{k})k_jJ^j(\mathbf{k}) - \mathcal{D}_a^0(\mathbf{k}',\mathbf{k})J^a(\mathbf{k})\,. \tag{83}$$

We now Fourier transform (76) and get:

$$E^i(\mathbf{k}') = \int d^4\mathbf{k}\left[ik'^i\,\mathcal{D}_\alpha^0(\mathbf{k}',\mathbf{k}) + \frac{i\omega}{c}\mathcal{D}_\alpha^i(\mathbf{k}',\mathbf{k})\right]J^\alpha(\mathbf{k})\,. \tag{84}$$

We can inject Eq. (82) and (83) in the latter expression and by substituting the summation index $a \to j$, we get the following:

$$E^i(\mathbf{k}') = \int d^3\mathbf{k}\,d\omega\left[\frac{i}{\omega}k'^i\,\mathcal{D}_0^0(\mathbf{k}',\mathbf{k})k_j - \frac{i}{c}k'^i\,\mathcal{D}_j^0(\mathbf{k}',\mathbf{k}) + \frac{i}{c}\mathcal{D}_0^i(\mathbf{k}',\mathbf{k})k_j - \frac{i\omega}{c^2}\mathcal{D}_j^i(\mathbf{k}',\mathbf{k})\right]J^j(\mathbf{k})\,, \tag{85}$$

where we used $d^4\mathbf{k} = d^3\mathbf{k}\,d\omega/c$. A Fourier transform with respect to $\mathbf{r}$ and $\mathbf{r}'$ then gives:

$$\begin{aligned} E^i(\mathbf{r}',\omega) = \int d^3\mathbf{r}\,d\omega\bigg[&-\frac{i}{\omega}\nabla'^i\,\mathcal{D}_0^0(\mathbf{r}',\mathbf{r},\omega)\nabla_j - \frac{1}{c}\nabla'^i\,\mathcal{D}_j^0(\mathbf{r}',\mathbf{r},\omega) + \frac{1}{c}\mathcal{D}_0^i(\mathbf{r}',\mathbf{r},\omega)\nabla_j \\ &- \frac{i\omega}{c^2}\mathcal{D}_j^i(\mathbf{r}',\mathbf{r},\omega)\bigg]J^j(\mathbf{r},\omega)\,. \end{aligned} \tag{86}$$

Besides, for any fields $\Phi$ and $\mathbf{V}$ the integration by part in $\mathbb{R}^3$ reads:

$$\int_\Omega \Phi\boldsymbol{\nabla}.\mathbf{V}d\mathbf{r} = \int_{\partial\Omega} \Phi\mathbf{V}.d\mathbf{s} - \int_\Omega \mathbf{V}.\boldsymbol{\nabla}\Phi d\mathbf{r}\,, \tag{87}$$

which can be applied to the latter equation in order to get:

$$\begin{aligned} E^i(\mathbf{r}',\omega) = \int d^3\mathbf{r}\,d\omega\bigg[&\frac{i}{\omega}\nabla'^i\nabla_j\,\mathcal{D}_0^0(\mathbf{r}',\mathbf{r},\omega) - \frac{1}{c}\nabla'^i\,\mathcal{D}_j^0(\mathbf{r}',\mathbf{r},\omega) - \frac{1}{c}\nabla_j\mathcal{D}_0^i(\mathbf{r}',\mathbf{r},\omega) \\ &- \frac{i\omega}{c^2}\mathcal{D}_j^i(\mathbf{r}',\mathbf{r},\omega)\bigg]J^j(\mathbf{r},\omega)\,. \end{aligned} \tag{88}$$

We can now use the definition (8) in order to identify the screened Green dyadic and get:

$$\begin{aligned} \mathcal{G}_j^i(\mathbf{r}',\mathbf{r},\omega) = &\frac{-1}{4\pi\omega^2}\nabla'^i\nabla_j\,\mathcal{D}_0^0(\mathbf{r}',\mathbf{r},\omega) + \frac{i}{4\pi\omega c}\left[\nabla'^i\,\mathcal{D}_j^0(\mathbf{r}',\mathbf{r},\omega) + \nabla_j\mathcal{D}_0^i(\mathbf{r}',\mathbf{r},\omega)\right] \\ &+ \frac{1}{4\pi c^2}\mathcal{D}_j^i(\mathbf{r}',\mathbf{r},\omega)\,. \end{aligned} \tag{89}$$

To the best of our knowledge, this equation has never been derived so far. We can also Fourier transform it back with respect to $\mathbf{r}$ and $\mathbf{r}'$ and get:

$$\begin{aligned} \mathcal{G}_j^i(\mathbf{k}',\mathbf{k},\omega) = &\frac{1}{4\pi\omega^2}k'^i k_j\,\mathcal{D}_0^0(\mathbf{k}',\mathbf{k},\omega) - \frac{1}{4\pi\omega c}\left[k'^i\,\mathcal{D}_j^0(\mathbf{k}',\mathbf{k},\omega) + k_j\mathcal{D}_0^i(\mathbf{k}',\mathbf{k},\omega)\right] \\ &+ \frac{1}{4\pi c^2}\mathcal{D}_j^i(\mathbf{k}',\mathbf{k},\omega)\,. \end{aligned} \tag{90}$$

### 5.2.3 Reciprocity theorem and symmetry properties of the Green dyadic

We now impose the following condition:

$$\mathscr{S}\left(\mathcal{G}_j^i(\boldsymbol{r}',\boldsymbol{r},\omega)\right) = \mathcal{G}_i^j(\boldsymbol{r},\boldsymbol{r}',\omega) = \mathcal{G}_j^i(\boldsymbol{r}',\boldsymbol{r},\omega), \tag{91}$$

where we define the operator $\mathscr{S}$ exchanging the indexes $i \leftrightarrow j$ and coordinates $\boldsymbol{r} \leftrightarrow \boldsymbol{r}'$. This *reciprocity* condition [11] states that a current in $\boldsymbol{r}$ creating a EM field in $\boldsymbol{r}'$ is equivalent to a current in $\boldsymbol{r}'$ creating a EM field in $\boldsymbol{r}$. In some particular situations (e.g. chiral meta-materials, moving media or topological materials), the latter condition is no longer true [85]. In this work, we only consider reciprocal media.

Therefore, in a *reciprocal medium*, the Green dyadic imposes (see appendix B for the detailed derivation):

$$\mathcal{G}_j^i(\boldsymbol{r}',\boldsymbol{r},\omega) = \frac{1}{4\pi c^2}\mathcal{D}_j^i(\boldsymbol{r}',\boldsymbol{r},\omega) - \frac{1}{4\pi\omega^2}\nabla'^i\nabla_j\mathcal{D}_0^0(\boldsymbol{r}',\boldsymbol{r},\omega), \tag{92}$$

where the first term is a charge-charge correlator while the second term is a current-current correlator. Therefore, the Green dyadic can be written, with the susceptibility tensor components expressed in the Lorenz gauge, as:

$$\mathcal{G}_j^i(\boldsymbol{k}',\boldsymbol{k},\omega) = \frac{4\pi}{k^2 - \frac{\omega^2}{c^2}}\left(\frac{1}{4\pi\omega^2}k'^i k_j \chi_0^0(\boldsymbol{k}',\boldsymbol{k},\omega) + \frac{1}{4\pi c^2}\chi_j^i(\boldsymbol{k}',\boldsymbol{k},\omega)\right)\frac{4\pi}{k'^2 - \frac{\omega^2}{c^2}}. \tag{93}$$

This equation is probably the most important new result of the section as it generalises the Kubo approach derived in the quasi-static case (59) to the retarded regime. Indeed by taking the quasi-static limit $c \longrightarrow \infty$ we obtain:

$$\mathcal{G}_j^i(\boldsymbol{k}',\boldsymbol{k},\omega) = \frac{4\pi}{\omega^2}\frac{k_j k'^i}{k^2 k'^2}\chi_0^0(\boldsymbol{k}',\boldsymbol{k},\omega), \tag{94}$$

which corresponds to the formula (59) that we derived in the quasi-static formalism with $\chi = \chi_0^0$. Upon insertion of the screened Green's function (93) into the loss probability (5) we finally obtain a fully retarded version the electron energy loss probability

$$\Gamma_{i\to f}^{R} = -\frac{4}{\pi}\sum_f \iint d\boldsymbol{r}\,d\boldsymbol{r}'J_{i\to f}(\boldsymbol{r})\text{Im}\{\overleftrightarrow{\mathcal{G}}(\boldsymbol{r},\boldsymbol{r}',\omega)\}J_{i\to f}^*(\boldsymbol{r})]\delta(\epsilon_f - \epsilon_i + \hbar\omega). \tag{95}$$

An alternative derivation of $\Gamma^R$ is based on inverting (92) using the divergence-free property of the susceptibility [73]. The obtained $\mathcal{D}$ is then inserted in (47). Finally the charge component is replaced using the continuity equation (79).

Eq. (95) may be transformed to Abajo's expression Eq. (11) by replacing the screened Green's dyad Eq. (90) with the semiclassical one and noting that the transition currents read

$$\begin{aligned} J_{i\to f}(\boldsymbol{r}) &\approx 2ik_z\psi_f(\boldsymbol{r})\psi_i^*(\boldsymbol{r}) \\ &\approx 2\psi_f(\boldsymbol{r})\boldsymbol{\nabla}\psi_i^*(\boldsymbol{r}) \end{aligned} \tag{96}$$

in the paraxial case. Finally, by combining Eq. (45) and (92), we can directly connect the mutual coherence tensor to the relativistic MDFF as:

$$\begin{aligned} \Lambda_j^i(\boldsymbol{r}',\boldsymbol{r},\omega) &= \frac{1}{4\pi\omega^2}\mathscr{F}_{\boldsymbol{k},-\boldsymbol{k}'}\left[\frac{\boldsymbol{k}\boldsymbol{k}'\,S_0^0(\boldsymbol{k}',\boldsymbol{k},\omega)}{(k^2-\omega^2)(k^2-\omega^2)}\right] \\ &\quad -\frac{1}{4\pi c^2}\mathscr{F}_{\boldsymbol{k},-\boldsymbol{k}'}\left[\frac{S_j^i(\boldsymbol{k}',\boldsymbol{k},\omega)}{(k^2-\omega^2)(k^2-\omega^2)}\right]. \end{aligned} \tag{97}$$

## 5.3   Concluding remarks

In the nano-optics framework, one usually calculates the Green dyadic $\overleftrightarrow{G}$ as this is the sole quantity required to describe the equilibrium properties of the electromagnetic field as demonstrated by Agarwal [84] and recalled in Sec. 3.2. On the other hand, from a condensed matter physicist point-of-view, the relevant quantity is the mixed dynamic form factor (or the susceptibility) of the material because it encodes all the information on the space-time dependent electronic correlations as demonstrated by Van Hove [86,87].

These two approaches are completely equivalent and based on the fluctuation-dissipation theorem which connects the response of the system (Green dyadic or susceptibility) to correlations of the underlying fields (electromagnetic or electronic correlation). The essence of Sec. 5 was to explicitly show this equivalence and demonstrate that, to the first order, the two propagators $\chi$ and $W$ (or $\chi^{\nu}_{\mu}$ and $G^{j}_{i}$) are simply connected by two vacuum photon propagators.

## 6   Kinetic equation for the electron density matrix

Following the logic of diagram (3), we will focus on the electron probe. Lets consider a fast electron described by the wave-function $\psi(\boldsymbol{r}, t)$. We can then define the *single electron density matrix* as:

$$\rho(\boldsymbol{r}, t, \boldsymbol{r}', t') = \psi(\boldsymbol{r}, t)\psi^*(\boldsymbol{r}', t'). \tag{98}$$

In the following, we will also consider the case of a density matrix invariant by translation in time $\rho(\boldsymbol{r}, t, \boldsymbol{r}', t') = \rho(\boldsymbol{r}, \boldsymbol{r}', t - t')$. In this case, the corresponding Fourier transform reads:

$$\rho(\boldsymbol{r}, \boldsymbol{r}', \omega) = \frac{1}{2\pi} \int d(t - t') \rho(\boldsymbol{r}, \boldsymbol{r}', t - t') e^{i\omega(t - t')}. \tag{99}$$

If the Hamiltonian is time independent then the corresponding wavefunction becomes separable $\psi(\boldsymbol{r}, t) = \psi(\boldsymbol{r})e^{i\epsilon t}$ and the density matrix can be written:

$$\rho(\boldsymbol{r}, \boldsymbol{r}', \omega) = \psi(\boldsymbol{r})\psi(\boldsymbol{r}')\delta(\omega - \epsilon), \tag{100}$$

which corresponds to the already defined above spectral one-electron density matrix. The goal of this section is to calculate the kinetic equation for the density matrix i.e. the equation ruling the evolution of the density matrix during elastic and inelastic scattering events. In the quasi-static limit, this equation has been derived for the first time by Dudarev, Peng and Whelan [26] and reads:

$$\rho_f(\boldsymbol{r}, \boldsymbol{r}', E) = \int d\boldsymbol{r}_1 d\boldsymbol{r}'_1 \, U_0(\boldsymbol{r}, \boldsymbol{r}_1, E) \, U_0^{\dagger}(\boldsymbol{r}', \boldsymbol{r}'_1, E) \mathscr{F}_{\boldsymbol{r}_1, -\boldsymbol{r}'_1} \left\{ \frac{S(\boldsymbol{k}, \boldsymbol{k}', \omega)}{k^2 k'^2} \right\} \rho_i(\boldsymbol{r}_1, \boldsymbol{r}'_1, E + \hbar\omega), \tag{101}$$

where $U_0$ is the time evolution operator of the free electron. This equation has then been introduced to EELS by Schattschneider and collaborators [15] and later applied to various situations such as EMCD [88], core-loss spectroscopy [22] or diffraction [89]. The goal of this section is to adapt this formula to the case of a retarded interaction kernel. As a matter of fact, apart from the final step, the derivation is essentially the same both in the quasi-static and the retarded case. Therefore, this section is organized as follows. In Sec. 6.1, 6.2 and 6.3, we review the seminal demonstration of Dudarev and collaborators with special emphasis on the different approximations made. The result of the derivation is a kinetic equation in the temporal domain valid for any weak interaction $V$. In Sec. 6.4, we use an explicit expression

for $V$ and derive the kinetic equation in the spectral domain in both the quasi-static and retarded interactions case. To do so, we use the result of Sec. 5 and assume a steady-state of illumination for the electron beam.

## 6.1 Schrödinger equation for the electron propagator

Without loss of generality, we consider a fast electron interacting with a target. The Hamiltonian of the total system {target + $e^-$} is then given by:

$$\hat{H}_{\text{tot}} = \hat{H}_{\text{t}} + \hat{H}_e + \hat{H}_{\text{int}}, \tag{102}$$

where $\hat{H}_e$ describes the free propagation of the electron, $\hat{H}_{\text{t}}$ encodes the electronic properties of the target only and $\hat{H}_{\text{int}}$ gives the interaction between the excitations in the target and the impinging electron. We now separate the interaction potential into its thermodynamical average and a fluctuating part:

$$\hat{H}_{\text{int}} = \langle \hat{H}_{\text{int}} \rangle + \hat{V}. \tag{103}$$

The thermodynamical average is taken over the ensemble of realizations of target states:

$$\langle \hat{H}_{\text{int}} \rangle = \frac{1}{Z} \sum_n \langle n | \hat{H}_{\text{int}} | n \rangle \, e^{-\beta \epsilon_n} \tag{104}$$

employing the same notations we used in Sec. 5. This term now encompasses elastic scattering and static field contribution. To simplify the notation, we suppose that $\langle \hat{H}_{\text{int}} \rangle = 0$ which has no consequence on the following derivation. A non vanishing average could be absorbed by modifying the free electron Hamiltonian as:

$$\hat{H}'_e = H_e + \langle \hat{H}_{\text{int}} \rangle = -\frac{\hbar^2}{2m} \nabla^2 + \langle \hat{H}_{\text{int}} \rangle. \tag{105}$$

Besides, the time evolution operator $\hat{U}_0$ of the free electron as well as the time evolution operator for the total system $\hat{T}$ follow the Schrödinger equation:

$$\begin{cases} i \dfrac{\partial}{\partial t} \hat{U}_0(t, t_0) = \hat{H}_e \, \hat{U}_0(t, t_0) + \delta(t - t_0) & \text{(106a)} \\[2mm] i \dfrac{\partial}{\partial t} \hat{T}(t, t_0) = \hat{H}_{\text{tot}} \, \hat{T}(t, t_0) + \delta(t - t_0). & \text{(106b)} \end{cases}$$

Eq. (106a) can be straightforwardly integrated and gives:

$$\hat{U}_0(t - t_0) = -i\Theta(t - t_0)e^{-i\hat{H}_e(t - t_0)}. \tag{107}$$

However, Eq. (106b) cannot be explicitly solved. To overcome this difficulty, we first define the operator $\hat{U}$ as:

$$\hat{U}(t, t_0) = e^{i\hat{H}_{\text{t}}(t - t_0)}\hat{T}(t, t_0), \tag{108}$$

which corresponds to the evolution operator for the *interacting* electron. Moreover, we define the Heisenberg representation of the fluctuating part of the interaction $\hat{V}$ as:

$$\hat{V}(t - t_0) = e^{i\hat{H}_{\text{t}}(t - t_0)}\hat{V}e^{-i\hat{H}_{\text{t}}(t - t_0)}. \tag{109}$$

Combining Eq. (106b), (108) and (109), we get the Schrödinger equation for the time evolution operator of the interacting electron:

$$i\frac{\partial \hat{U}(t, t_0)}{\partial t} = \left(\hat{H}_e + \hat{V}(t - t_0)\right)\hat{U}(t, t_0) + \delta(t - t_0). \tag{110}$$

In the next section, we will use a perturbation approach in order to calculate a good approximation of this evolution operator.

## 6.2 Dyson equation for the single electron propagator

We first integrate Eq. (110) in order to obtain the following integral representation:

$$\hat{U}(t,t_0) = \hat{U}_0(t,t_0) + \frac{1}{i}\int_{t_0}^{t} dt_1 \hat{U}_0(t,t_1)\hat{V}(t_1)\hat{U}(t_1,t_0). \tag{111}$$

The latter equation can be solved iteratively by writing:

$$\hat{U}(t,t_0) = \hat{U}_0(t,t_0) + \frac{1}{i}\int_{t_0}^{t} dt_1 \hat{U}_0(t,t_1)\hat{V}(t_1)\hat{U}_0(t_1,t_0)$$
$$+ \frac{1}{i^2}\int_{t_0}^{t} dt_1 \hat{U}_0(t,t_1)\hat{V}(t_1)\int_{t_0}^{t_1} dt_2 \hat{U}_0(t_1,t_2)\hat{V}(t_2)\hat{U}_0(t_2,t_0) + \dots. \tag{112}$$

The latter equation can be diagrammatically schematized as:

$$\tag{113}$$

Let's now re-arrange the previous integrals by looking at the second term in (112) and for brevity taking $\hat{U}_0 = \mathrm{Id}$, where Id denotes the identity operator. Separating the integral in two parts and changing the integration variable leads to:

$$\int_{t_0}^{t} dt_1 \hat{V}(t_1)\int_{t_0}^{t_1} dt_2 \hat{V}(t_2) = \frac{1}{2}\int_{t_0}^{t} dt_1 \hat{V}(t_1)\int_{t_0}^{t_1} dt_2 \hat{V}(t_2)$$
$$+ \frac{1}{2}\int_{t_0}^{t} dt_2 \hat{V}(t_2)\int_{t_0}^{t_2} dt_1 \hat{V}(t_1). \tag{114}$$

The integration limit of the integrals can then be all set to $t_0$ and $t$ if one introduces the proper Heaviside functions:

$$\int_{t_0}^{t} dt_1 \hat{V}(t_1)\int_{t_0}^{t_1} dt_2 \hat{V}(t_2) = \frac{1}{2}\int_{t_0}^{t} dt_1 \int_{t_0}^{t} dt_2\, \hat{V}(t_1)\hat{V}(t_2)$$
$$\times \theta(t_1 - t_2) + \frac{1}{2}\int_{t_0}^{t} dt_2 \int_{t_0}^{t} dt_1\, \hat{V}(t_2)\hat{V}(t_1)\theta(t_2 - t_1). \tag{115}$$

And using the definition of the time ordering operator (188), we finally obtain:

$$\int_{t_0}^{t} dt_1 \hat{V}(t_1)\int_{t_0}^{t_1} dt_2 \hat{V}(t_2) = \frac{1}{2}\int_{t_0}^{t} dt_1 \int_{t_0}^{t} dt_2\, \mathcal{T}\{\hat{V}(t_1)\hat{V}(t_2)\}. \tag{116}$$

The same trick can be applied to all order but keeping in mind that the prefactor for the $n^{\text{th}}$ order term is $(1/n!)$. It enables to re-write Eq. (112) as:

$$\hat{U}(t,t_0) = \sum_{n=0}^{\infty} \frac{(-i)^n}{n!} \int_{t_0}^{t} dt_1 \dots \int_{t_0}^{t} dt_n \mathcal{T}\{\hat{U}_0(t,t_1)\hat{V}(t_1) \\ \times \hat{U}_0(t_1,t_2)\dots\hat{V}(t_n)\hat{U}_0(t_n,t_0)\}. \tag{117}$$

In order to use the linear response theory derived in Sec. (5), we now calculate the average value of the exact electron propagator $\langle \hat{U}(t,t_0) \rangle \equiv \hat{\mathcal{U}}(t,t_0)$:

$$\hat{\mathcal{U}}(t,t_0) = \sum_{n=0}^{\infty} \frac{(-i)^n}{n!} \int_{t_0}^{t} dt_1 \dots \int_{t_0}^{t} dt_n \Big\langle \mathcal{T}\{\hat{U}_0(t,t_1)\hat{V}(t_1) \\ \times \hat{U}_0(t_1,t_2)\dots\hat{V}(t_n)\hat{U}_0(t_n,t_0)\}\Big\rangle. \tag{118}$$

We now use the Isserlis-Wick theorem which states that for a set of Gaussian random variables $\{X_1,\dots,X_n\}$, any monomial of these variables satisfies:

$$\langle X_1 X_2 \dots X_{2m+1} \rangle = 0 \tag{119a}$$

$$\langle X_1 X_2 \dots X_{2m} \rangle = \sum_{\substack{\text{All possible} \\ \text{pairings}}} \prod_{i,j} \text{Cov}[X_i X_j], \tag{119b}$$

where Cov denotes the covariance. And since by construction the mean value of $\hat{V}$ is zero, we have $\text{Cov}[V(t_i)V(t_j)] = \langle V(t_i)V(t_j) \rangle - \langle V(t_i) \rangle \langle V(t_j) \rangle = \langle V(t_i)V(t_j) \rangle$. Eq. (118) then becomes:

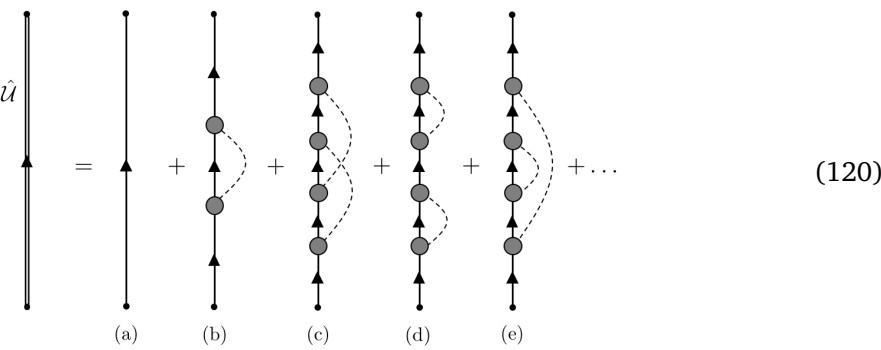

$$\tag{120}$$

where each dotted line represents a covariance product $\langle V(t_i)V(t_j) \rangle$. We now turn to the main approximation of this development [90–92]: *we only keep diagrams involving correlations between neighboring vertexes*. For example, we therefore neglect terms (c) and (e) in Eq. (120). This approximation can be interpreted in two equivalent ways:

• First, as pointed out in [26], this approximation consists in treating all the successive scattering events as single independent events which correspond to the Born approximation. Following [26,93], to determine its condition of validity, we introduce the typical correlation length $r_c$ of the excitations in the target, $v$ the speed of the traveling electron and $|\hat{V}|$ the order of magnitude of the interaction. Then this approximation holds if:

$$\frac{\hbar v}{r_c} \gg |\hat{V}| \tag{121}$$

In other terms, the correlation length should be short enough, or the interaction weak enough, for no dynamical effects to appear. Nevertheless, this Born approximation applies to the fluctuating part of the interaction only while the static part is included *a priori*. Thus, this approximation is rather a distorted-wave Born approximation [26].

• One can also interpret this approximation in a quantum field theory fashion [94] as the dotted lines can be regarded as a particle exchange. In this case, the approximation above consists in not allowing two (or more) simultaneous excitations, which is valid in the weak interaction limit. We exemplify it on the following diagram:

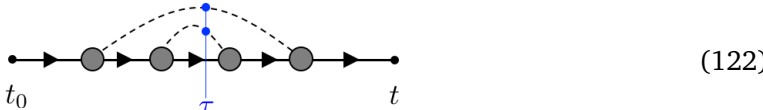

$$\text{(122)}$$

Thanks to the approximation made, Eq. (120) is dramatically simplified and can be factorized as follow:

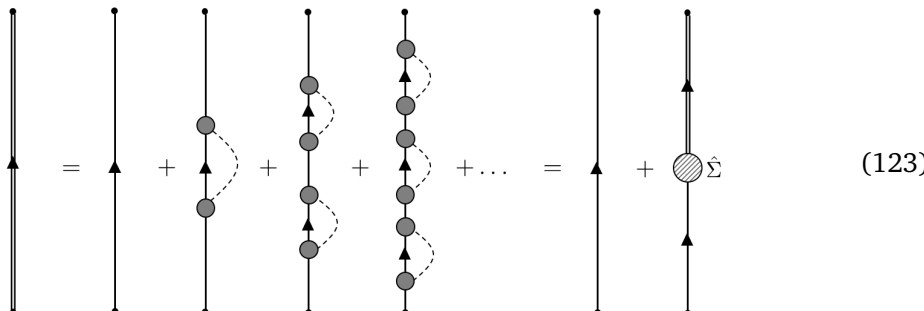

$$\text{(123)}$$

where $\hat{\Sigma}$ is the *self-energy* of the probe electrons [95] and reads, in a synthetic form:

$$\hat{\Sigma} = \hat{U}_0^{-1} - \hat{\mathcal{U}}^{-1} \approx \langle \mathcal{T}\{\hat{V}\hat{U}_0\hat{V}\}\rangle . \tag{124}$$

Eq. (124) is the starting point of Echenique's et al. formalism [54] that we will review at the end of this section. The Dyson equation (123) can be re-written in its explicit form, in the time domain, as:

$$\hat{\mathcal{U}}(t,t_0) = \hat{U}_0(t,t_0) - \int_{t_0}^t dt_1 \int_{t_0}^t dt_2 \, \hat{U}_0(t,t_1)\langle \mathcal{T}\{\hat{V}(t_1)U_0(t_1,t_2)\hat{V}(t_2)\}\rangle\hat{\mathcal{U}}(t_2,t_0). \tag{125}$$

## 6.3 Bi-linear propagator for the single electron density matrix

We will construct the propagator of the single-electron density matrix. To do so, we will use (125) to construct an average propagator $\hat{\mathcal{K}}$ of the exact density-matrix propagator $\hat{K}$. Starting from the exact electron propagator $\hat{U}$, one can construct $\hat{K}$ as a tensor product:

$$\hat{K} = \hat{U} \otimes \hat{U}^\dagger . \tag{126}$$

Injecting the development (117) in the latter development, we obtain:

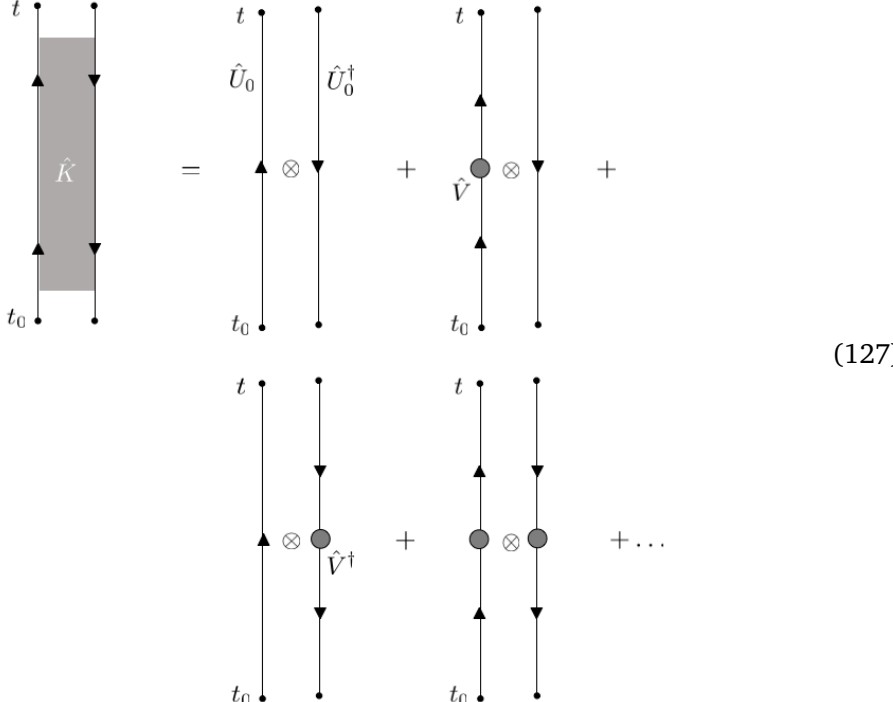

$$(127)$$

In the following, for brevity reasons, we will omit the $\otimes$ symbol in the diagrams. As we did for the electron propagator, we now take the average value of $\hat{K}$. Using the Isserlis-Wick theorem, we obtain the following expression for $\hat{\mathcal{K}}$:

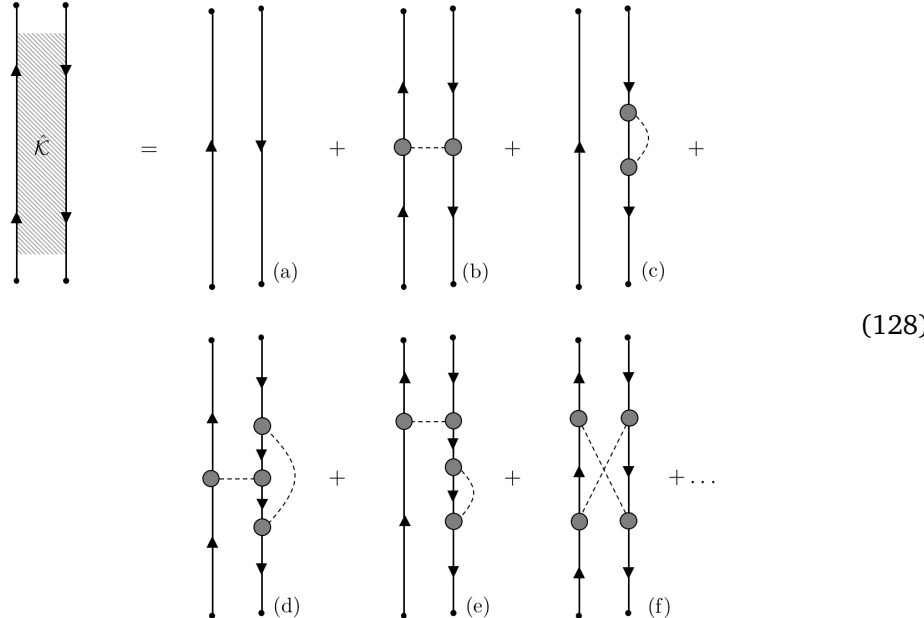

$$(128)$$

At this point, we will make the same approximation as in the last section and neglect all diagrams with several simultaneous excitations, e.g., diagram (d) in (128). Diagrams like (f) correspond to coherent back-scattering events which are typically sufficiently small to be neglected [26]. This approximation is the so-called forward scattering approximation and is standard in electron microscopy.

Within these approximations, the expansion contains only two building blocks: the electron self-energy term (c) and mutual correlations (b). We can of course encounter sequences of these blocks like diagram (e). We can then partially re-sum the self-energy terms which leads to:

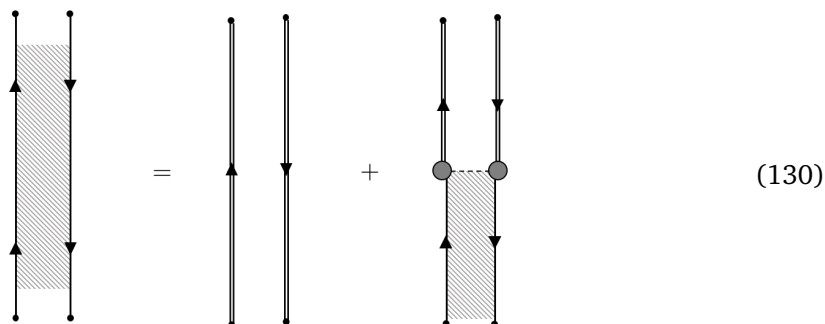

$$(129)$$

The latter equation formally corresponds to a Bethe-Salpeter equation in the very specific case where the two bound states correspond to $\psi$ and $\psi^\dagger$ and within the so-called *ladder approximation*. This equation can be re-summed and reads:

$$(130)$$

Or in its explicit form:

$$
\begin{aligned}
\hat{\mathcal{K}}(\boldsymbol{r},t;\boldsymbol{r}_0,t_0|\boldsymbol{r}',t';\boldsymbol{r}'_0,t'_0) =& \hat{\mathcal{U}}(\boldsymbol{r},t;\boldsymbol{r}_0,t_0)\,\hat{\mathcal{U}}^\dagger(\boldsymbol{r}',t';\boldsymbol{r}'_0,t'_0) \\
&+ \int_{t_0}^{t} dt_1 dt'_1 \int_{t_0}^{t} d\boldsymbol{r}_1 d\boldsymbol{r}'_1\, \hat{\mathcal{U}}(\boldsymbol{r},t;\boldsymbol{r}_1,t_1)\hat{\mathcal{U}}^\dagger(\boldsymbol{r}',t';\boldsymbol{r}'_1,t'_1) \\
&\times \left\langle \mathcal{T}\{\hat{V}(\boldsymbol{r}_1,t_1)\hat{V}^\dagger(\boldsymbol{r}'_1,t'_1)\}\right\rangle \hat{\mathcal{K}}(\boldsymbol{r}_1,t_1;\boldsymbol{r}_0,t_0|\boldsymbol{r}'_1,t'_1;\boldsymbol{r}'_0,t'_0).
\end{aligned}
\tag{131}
$$

## 6.4 The kinetic equation for the single electron density matrix

We are now in position to derive the master equation describing the propagation of the single electron density matrix, i.e., the so-called *kinetic equation*. Thus let's consider an incident density matrix $\rho_i(\boldsymbol{r}_0,t_0;\boldsymbol{r}'_0,t'_0)$ and propagate it to the point $(\boldsymbol{r},t;\boldsymbol{r}',t')$. Taking into account the interaction with the target and using the approximations detailed earlier, the final density matrix $\rho_f(\boldsymbol{r},t;\boldsymbol{r}',t')$ satisfies:

$$
\rho_f(\boldsymbol{r},t;\boldsymbol{r}',t') = \int dt_0\, dt'_0 \int d\boldsymbol{r}_0\, d\boldsymbol{r}'_0\, \hat{\mathcal{K}}(\boldsymbol{r},t;\boldsymbol{r}_0,t_0|\boldsymbol{r}',t';\boldsymbol{r}'_0,t'_0)\rho_i(\boldsymbol{r}_0,t_0;\boldsymbol{r}'_0,t'_0)
\tag{132}
$$

plugging (131) in the latter, we finally get:

$$\rho_f(\boldsymbol{r},t;\boldsymbol{r}',t') = \rho_0(\boldsymbol{r},t;\boldsymbol{r}',t') + \int dt_1 dt_1' \int d\boldsymbol{r}_1 d\boldsymbol{r}_1'$$
$$\mathcal{U}(\boldsymbol{r},t;\boldsymbol{r}_1,t_1)\mathcal{U}^\dagger(\boldsymbol{r}',t';\boldsymbol{r}_1',t_1')\hat{C}(\boldsymbol{r}_1,t_1,\boldsymbol{r}_1',t_1')\rho_i(\boldsymbol{r}_1,t_1;\boldsymbol{r}_1',t_1'),$$
(133)

where the correlation function reads $C(\boldsymbol{r}_1,t_1,\boldsymbol{r}_1',t_1') = \left\langle \mathcal{T}\{\hat{V}(\boldsymbol{r}_1,t_1)\hat{V}^\dagger(\boldsymbol{r}_1',t_1')\}\right\rangle$. Eq. (133) is the kinetic equation in the temporal domain where the interaction Hamiltonian is not yet specified. Let's highlight that at this point, the latter equation is very general and can be applied to model, e.g., time-resolved spectroscopy experiments.

We now suppose that the electron beam is in a *steady-state of illumination* which is the case in the standard EELS experiment we are describing here. In this case, the density matrix only depends on the time difference. We now Fourier transform Eq. (133) with respect to $t$ and $t'$ therefore taking the limits of the integrals over $t_1$ and $t_1'$ to be $\pm\infty$. We therefore obtain:

$$\rho_f(\boldsymbol{r},E,\boldsymbol{r}',E') = \rho_0(\boldsymbol{r},E,\boldsymbol{r}',E') + \int dt e^{-iEt} dt' e^{iE't'} \int dt_1 dt_1' \int d\boldsymbol{r}_1 d\boldsymbol{r}_1' \, \mathcal{U}(\boldsymbol{r},\boldsymbol{r}_1,t-t_1)$$
$$\times \mathcal{U}^\dagger(\boldsymbol{r}'\boldsymbol{r}_1',t'-t_1')\hat{C}(\boldsymbol{r}_1,\boldsymbol{r}_1',t_1-t_1')\,\rho_i(\boldsymbol{r}_1,\boldsymbol{r}_1',t_1-t_1').$$
(134)

Changing the integration variables leads to:

$$\rho_f(\boldsymbol{r},E,\boldsymbol{r}',E') = \rho_0(\boldsymbol{r},E,\boldsymbol{r}',E') + \int dt_1 dt_1' \int d\boldsymbol{r}_1 d\boldsymbol{r}_1' \hat{C}(\boldsymbol{r}_1,\boldsymbol{r}_1',t_1-t_1')\,\rho_i(\boldsymbol{r}_1,\boldsymbol{r}_1',t_1-t_1')$$
$$\times \int dt e^{-iE(t+t_1)} dt' e^{iE'(t'+t_1')} \mathcal{U}(\boldsymbol{r},\boldsymbol{r}_1,t)\mathcal{U}^\dagger(\boldsymbol{r}'\boldsymbol{r}_1',t'),$$
(135)

which can be re-written as:

$$\rho_f(\boldsymbol{r},E,\boldsymbol{r}',E') = \rho_0(\boldsymbol{r},E,\boldsymbol{r}',E') + \int d\boldsymbol{r}_1 d\boldsymbol{r}_1' \mathcal{U}(\boldsymbol{r},\boldsymbol{r}_1,E)\mathcal{U}^\dagger(\boldsymbol{r}'\boldsymbol{r}_1',E')$$
$$\times \int dt_1 dt_1' \hat{C}(\boldsymbol{r}_1,\boldsymbol{r}_1',t_1-t_1')\rho_i(\boldsymbol{r}_1,\boldsymbol{r}_1',t_1-t_1')e^{-iEt_1}e^{iE't_1'}.$$
(136)

And we recognize a convolution product with respect to $t_1-t_1'$. Noting $-\omega$ the convolution variable, we finally get:

$$\rho_f(\boldsymbol{r},\boldsymbol{r}',E) = \rho_0(\boldsymbol{r},\boldsymbol{r}',E) + \int d\boldsymbol{r}_1 d\boldsymbol{r}_1' \, \mathcal{U}(\boldsymbol{r},\boldsymbol{r}_1,E)\mathcal{U}^\dagger(\boldsymbol{r}'\boldsymbol{r}_1',E)$$
$$\times \int d\omega \, \hat{C}(\boldsymbol{r}_1,\boldsymbol{r}_1',-\omega)\,\rho_i(\boldsymbol{r}_1,\boldsymbol{r}_1',E+\omega).$$
(137)

We now need to calculate the Fourier transform of the correlation function. We will distinguish the quasi-static from the retarded case and denote the corresponding correlation functions with $\hat{C}^{QS}$ and $\hat{C}^R$, respectively.

### 6.4.1 First case: Quasistatic interaction kernel

The quasi-static interaction $\hat{V}^{\text{QS}}$ between the electron and the target is given by the Coulomb interaction:

$$\langle\psi_f|\hat{V}^{\text{QS}}(\boldsymbol{r},t)|\psi_i\rangle = \langle\psi_f|\int d\boldsymbol{r}'\frac{\hat{\text{n}}(\boldsymbol{r}',t)}{|\boldsymbol{r}-\boldsymbol{r}'|}|\psi_i\rangle\,, \tag{138}$$

where $\hat{\rho}$ is the charge density operator for the target. Therefore, $C^{\text{QS}}$ reads:

$$\hat{C}^{\text{QS}}(\boldsymbol{r}_1,t_1,\boldsymbol{r}_1',t_1') = \int d\boldsymbol{r}_2\int d\boldsymbol{r}_2'\frac{\langle 0|\mathcal{T}\{\hat{\text{n}}(\boldsymbol{r}_2,t_1)\hat{\text{n}}^\dagger(\boldsymbol{r}_2',t_1')\}|0\rangle}{|\boldsymbol{r}_1-\boldsymbol{r}_2||\boldsymbol{r}_1'-\boldsymbol{r}_2'|}\,. \tag{139}$$

Writing explicitly the time ordering operator, we get:

$$\hat{C}^{\text{QS}}(\boldsymbol{r}_1,\boldsymbol{r}_1',t_1-t_1') = \left[\int d\boldsymbol{r}_2\int d\boldsymbol{r}_2'\frac{\sum_n\langle 0|\hat{\text{n}}(\boldsymbol{r}_2)|n\rangle\langle n|\hat{\text{n}}^\dagger(\boldsymbol{r}_2')|0\rangle}{|\boldsymbol{r}_1-\boldsymbol{r}_2||\boldsymbol{r}_1'-\boldsymbol{r}_2'|}e^{-i(\omega_0-\omega_n)(t_1-t_1')}\theta(t_1-t_1')\right]$$
$$+\left[t_1\leftrightarrow t_1'\right]. \tag{140}$$

Writing $\tau = t_1 - t_1'$, the Fourier transform reads:

$$\int e^{-i\omega\tau}\hat{C}^{\text{QS}}(\boldsymbol{r}_1,\boldsymbol{r}_1',\tau)d\tau = \left[\int d\boldsymbol{r}_2\int d\boldsymbol{r}_2'\frac{\sum_n\langle 0|\hat{\text{n}}(\boldsymbol{r}_2)|n\rangle\langle n|\hat{\text{n}}^\dagger(\boldsymbol{r}_2')|0\rangle}{|\boldsymbol{r}_1-\boldsymbol{r}_2||\boldsymbol{r}_1'-\boldsymbol{r}_2'|}\int d\tau\,e^{-i(\omega+\omega_0-\omega_n)\tau}\theta(\tau)\right]$$
$$+\int e^{-i\omega\tau}\left[\tau\leftrightarrow-\tau\right]d\tau\,, \tag{141}$$

which gives:

$$\hat{C}^{\text{QS}}(\boldsymbol{r}_1,\boldsymbol{r}_1',\omega) = \left[\int d\boldsymbol{r}_2\int d\boldsymbol{r}_2'\frac{\sum_n\langle 0|\hat{\text{n}}(\boldsymbol{r}_2)|n\rangle\langle n|\hat{\text{n}}^\dagger(\boldsymbol{r}_2')|0\rangle}{|\boldsymbol{r}_1-\boldsymbol{r}_2||\boldsymbol{r}_1'-\boldsymbol{r}_2'|}\right]\left[\pi\delta(\omega+\omega_0-\omega_n)\right.$$
$$\left.-i\mathcal{P}\left(\frac{1}{\omega+\omega_0-\omega_n}\right)\right]+\left[\mathscr{F}\right]^*\,, \tag{142}$$

where $\mathcal{P}$ denotes the Cauchy principal value. Using the fact that for any complex number $z\in\mathbb{C}$ we have $z+z^*=2\text{Re}(z)$, we finally get:

$$\hat{C}^{\text{QS}}(\boldsymbol{r}_1,\boldsymbol{r}_1',\omega) = 2\pi\int d\boldsymbol{r}_2\int d\boldsymbol{r}_2'\frac{\sum_n\langle 0|\hat{\text{n}}(\boldsymbol{r}_2)|n\rangle\langle n|\hat{\text{n}}^\dagger(\boldsymbol{r}_2')|0\rangle}{|\boldsymbol{r}_1-\boldsymbol{r}_2||\boldsymbol{r}_1'-\boldsymbol{r}_2'|}\delta(\omega+\omega_0-\omega_n)\,. \tag{143}$$

From Eq. (60), one can see that:

$$\hat{C}^{\text{QS}}(\boldsymbol{r}_1,\boldsymbol{r}_1',-\omega) = \text{Im}\{-W(\boldsymbol{r}_1,\boldsymbol{r}_1',\omega)\}\,. \tag{144}$$

Therefore, plugging it in (137), we finally obtain:

$$\rho_f(\boldsymbol{r},\boldsymbol{r}',E) = \rho_0(\boldsymbol{r},\boldsymbol{r}',E) + \int d\boldsymbol{r}_1 d\boldsymbol{r}_1'\,\mathcal{U}(\boldsymbol{r},\boldsymbol{r}_1,E)\mathcal{U}^\dagger(\boldsymbol{r}'\boldsymbol{r}_1',E)$$
$$\times\int d\omega\,\text{Im}\{-W(\boldsymbol{r}_1,\boldsymbol{r}_1',\omega)\}\rho_i(\boldsymbol{r}_1,\boldsymbol{r}_1',E+\omega)\,, \tag{145}$$

which, thanks to Eq. (61), can also be identified to the result on Dudarev's paper (14).

### 6.4.2 Second case: Retarded interaction kernel

The retarded interaction $\hat{V}^{\mathrm{R}}$ between the electron and the target is given by the minimal coupling Hamiltonian:

$$
\langle\psi_f|\hat{V}^{\mathrm{R}}(\mathbf{r},t)|\psi_i\rangle = -\frac{e}{m}\langle\psi_f|\mathrm{A}^{\mu}(\mathbf{r},t)p_{\mu}|\psi_i\rangle \tag{146}
$$

$$
= -\frac{ie}{m}\langle\psi_f|\mathrm{A}^{\mu}(\mathbf{r},t)\partial_{\mu}|\psi_i\rangle, \tag{147}
$$

where $\mathrm{A}^{\mu}$ is the 4-potential associated with the excitations in the target and $p_{\mu}$ is the electron 4-momentum operator. Note furthermore that we neglected again the (diamagnetic) $e^2A^2$-term, which is of higher order in the perturbation expansion.

Moreover, within the linear response theory, the photon propagator can also be connected to the 4-potential correlation function which gives [63]:

$$
\mathcal{D}^{\nu}_{\mu}(\mathbf{r},\mathbf{r}',t,t') = -i\theta(t-t')\langle 0|[\mathrm{A}_{\mu}(\mathbf{r},t),\mathrm{A}^{\nu}(\mathbf{r}',t')]|0\rangle, \tag{148}
$$

where $\mathcal{D}$ is again the screened propagator of the EM field (taking into account the polarizability of the medium) which has been introduced in Sec. 4 and 5. By strict analogy with Eq. (52) and (54), we obtain:

$$
\mathrm{Im}\left\{-\mathcal{D}^{\nu}_{\mu}(\mathbf{r},\mathbf{r}',\omega)\right\} = 2\pi\sum_{n}\langle 0|\mathrm{A}_{\mu}(\mathbf{r})|n\rangle\langle n|\mathrm{A}^{\nu}(\mathbf{r}')|0\rangle\,\delta(\omega+\omega_n-\omega_0). \tag{149}
$$

The Fourier transform of $C^{\mathrm{R}}$ can be done in the exact same way as in the quasi-static case and leads to:

$$
\hat{C}^{\mathrm{R}}(\mathbf{r}_1,\mathbf{r}'_1,-\omega) = \mathrm{Im}\left\{-\mathcal{D}^{\nu}_{\mu}(\mathbf{r}_1,\mathbf{r}'_1,\omega)\right\}\partial^{\mu}\partial'_{\nu}. \tag{150}
$$

Thus, plugging it in (137), we finally obtain:

$$
\rho_f(\mathbf{r},\mathbf{r}',E) = \rho_0(\mathbf{r},\mathbf{r}',E) + \int d\mathbf{r}_1 d\mathbf{r}'_1\,\mathcal{U}(\mathbf{r},\mathbf{r}_1,E)\mathcal{U}^{\dagger}(\mathbf{r}'\mathbf{r}'_1,E)
$$
$$
\times\int d\omega\,\mathrm{Im}\left\{-\mathcal{D}^{\nu}_{\mu}(\mathbf{r}_1,\mathbf{r}'_1,\omega)\right\}\partial^{\mu}\partial'_{\nu}\rho_i(\mathbf{r}_1,\mathbf{r}'_1,E+\omega). \tag{151}
$$

Using Eq. (45), one can express the latter equation in terms of relativistic MDFF:

$$
\rho_f(\mathbf{r},\mathbf{r}',E) = \rho_0(\mathbf{r},\mathbf{r}',E) + \int d\mathbf{r}_1 d\mathbf{r}'_1\,\mathcal{U}(\mathbf{r},\mathbf{r}_1,E)\mathcal{U}^{\dagger}(\mathbf{r}'\mathbf{r}'_1,E)
$$
$$
\times\int d\omega\,\mathscr{F}_{\mathbf{k},-\mathbf{k}'}\left\{\frac{S_{\mu\nu}(\mathbf{k},\mathbf{k}',\omega)}{(k^2-\omega^2)(k'^2-\omega^2)}\right\}\rho_i(\mathbf{r}_1,\mathbf{r}'_1,E+\omega), \tag{152}
$$

we can now expand the sums over $\mu$ and $\nu$ with respect to the spatial and temporal coordinates as we did in Eq. (77) and therefore obtain four terms respectively involving $\mathcal{D}^0_0$, $\mathcal{D}^i_0$, $\mathcal{D}^0_j$ and $\mathcal{D}^i_j$. As we developed in Sec. 5.2.3 and appendix B, to the price of the hypothesis that the medium is reciprocal, one can neglect the anti-symmetric terms $\mathcal{D}^i_0$ and $\mathcal{D}^0_j$. This being so, we finally get:

$$
\rho_f(\mathbf{r},\mathbf{r}',E) = \rho_0(\mathbf{r},\mathbf{r}',E)
$$
$$
+\int d\mathbf{r}_1 d\mathbf{r}'_1\,\mathcal{U}(\mathbf{r},\mathbf{r}_1,E)\mathcal{U}^{\dagger}(\mathbf{r}'\mathbf{r}'_1,E)\int d\omega\,\mathrm{Im}\left\{-\mathcal{D}^0_0(\mathbf{r}_1,\mathbf{r}'_1,\omega)\right\}\partial^0\partial'_0\,\rho_i(\mathbf{r}_1,\mathbf{r}'_1,E+\omega)
$$
$$
+\int d\mathbf{r}_1 d\mathbf{r}'_1\,\mathcal{U}(\mathbf{r},\mathbf{r}_1,E)\mathcal{U}^{\dagger}(\mathbf{r}'\mathbf{r}'_1,E)\int d\omega\,\mathrm{Im}\left\{\mathcal{D}^j_i(\mathbf{r}_1,\mathbf{r}'_1,\omega)\right\}\partial^i\partial'_j\,\rho_i(\mathbf{r}_1,\mathbf{r}'_1,E+\omega). \tag{153}
$$

The second term is a Coulomb term analogue to the third term in (145) while the second one is the retarded part of the interaction.

We now move to the temporal gauge $\phi = 0$ where, as we explained in Sec. 3.1 and is detailed in [62], the temporal part of the vacuum photon propagator cancels $D_0^0 = D_j^0 = D_0^i = 0$. Using the Dyson developments (73) and the expression of the dyadic Green function (89), Eq. (151) can be directly reduced to:

$$\rho_f(\mathbf{r}, \mathbf{r}', E) = \rho_0(\mathbf{r}, \mathbf{r}', E) + \int d\mathbf{r}_1 d\mathbf{r}_1' \, \mathcal{U}(\mathbf{r}, \mathbf{r}_1, E) \mathcal{U}^\dagger(\mathbf{r}' \mathbf{r}_1', E)$$
$$\times \int d\omega \, \overset{\leftrightarrow}{\Lambda}(\mathbf{r}_1, \mathbf{r}_1', \omega) \, \boldsymbol{\nabla} \boldsymbol{\nabla}' \rho_i(\mathbf{r}_1, \mathbf{r}_1', E + \omega). \tag{154}$$

All the quantities involved in the latter equation being gauge-independent, expression (154) must be valid in the general case of arbitrary gauge. Eq. (145), (151) and (154) are the essential results of this section. Before concluding, we will apply them to the case of electron energy loss spectroscopy.

# 7 Single scattering approximation: application to electron energy loss experiments

We now apply the previous result to the specific case of inelastic energy loss spectroscopy. We will therefore make further approximations:

1. The first term of the right hand side of Eq. (137) describes the elastic part of the interaction. As we are going to discuss EELS experiments in the following, we will not consider this term.

2. As done by Schattschneider, Nelhiebel and Jouffrey [15], we will consider a monochromatic electron, of energy $\epsilon_0$ and density matrix $\rho_i$, interacting a single time with the sample. It enables us to replace $\mathcal{U}$ by the free space electron Green's functions $U_0$.

3. As we are now interested in energy-resolved quantity, we remove the integral over $\omega$.

Under these assumptions Eq. (137) reads:

$$\rho_f(\mathbf{r}, \mathbf{r}', \epsilon_f) = \int d\mathbf{x} \, d\mathbf{x}' \, U_0(\mathbf{r}, \mathbf{x}, \epsilon_f) \, U_0^*(\mathbf{r}', \mathbf{x}', \epsilon_f) \hat{C}(\mathbf{x}, \mathbf{x}', \omega) \, \rho_i(\mathbf{x}, \mathbf{x}', \epsilon_f + \omega), \tag{155}$$

where we intentionally do not specify the operator $\hat{C}$ in order not to lose generality as both the quasi-static (144) and retarded (151) interactions can be used indifferently.

## 7.1 Electron energy loss probability

From Eq. (155), one can deduce the wave-optical EELS probability (9) and (11). To do so, we first decompose the final density matrix as (13):

$$\rho_f(\mathbf{r}, \mathbf{r}', \epsilon_f) = \sum_n p_n \psi_n(\mathbf{r}) \psi_n^*(\mathbf{r}') \delta(\epsilon_n - \epsilon_f), \tag{156}$$

while the initial electron can be considered as a monochromatic pure state [15] i.e.:

$$\rho_i(\boldsymbol{x}, \boldsymbol{x}', \epsilon_0) = \psi_i(\boldsymbol{x})\psi_i^*(\boldsymbol{x}')\delta(\epsilon_i - \epsilon_f - \omega). \tag{157}$$

We multiply each side of Eq. (155) by $\psi_n^*(\boldsymbol{r})\psi_n(\boldsymbol{r}')$, which leads to:

$$\rho_f(\boldsymbol{r}, \boldsymbol{r}', \epsilon_f)\psi_n^*(\boldsymbol{r})\psi_n(\boldsymbol{r}') = \int d\boldsymbol{x}\, d\boldsymbol{x}'\, U_0(\boldsymbol{r}, \boldsymbol{x}, \epsilon_f)U_0^*(\boldsymbol{r}', \boldsymbol{x}', \epsilon_f)$$
$$\times \hat{C}(\boldsymbol{x}, \boldsymbol{x}', \omega)\rho_i(\boldsymbol{x}, \boldsymbol{x}', \epsilon_0)\psi_n^*(\boldsymbol{r})\psi_n(\boldsymbol{r}')\delta(\epsilon_i - \epsilon_f - \omega). \tag{158}$$

We now perform an integral over $\boldsymbol{r}$ and $\boldsymbol{r}'$ which leads to:

$$\int d\boldsymbol{r}\, d\boldsymbol{r}'\, \rho(\boldsymbol{r}, \boldsymbol{r}', \epsilon_f)\psi_n^*(\boldsymbol{r})\psi_n(\boldsymbol{r}') = \int d\boldsymbol{x}\, d\boldsymbol{x}' \left[\int d\boldsymbol{r}\, U_0(\boldsymbol{r}, \boldsymbol{x}, \epsilon_f)\psi_n^*(\boldsymbol{r})\right]$$
$$\left[\int d\boldsymbol{r}\, U_0^*(\boldsymbol{r}', \boldsymbol{x}', \epsilon_f)\psi_n(\boldsymbol{r}')\right]\hat{C}(\boldsymbol{x}, \boldsymbol{x}', \omega)\psi_i(\boldsymbol{x})\psi_i^*(\boldsymbol{x}')\delta(\epsilon_i - \epsilon_f - \omega). \tag{159}$$

Since the Green function $U_0$ is symmetric with respect to the positions $\boldsymbol{x}$ and $\boldsymbol{r}$, we have by definition of the electron propagator:

$$\int d\boldsymbol{r}\, U_0(\boldsymbol{r}, \boldsymbol{x}, \epsilon_f)\psi_n^*(\boldsymbol{r}) = \psi_n^*(\boldsymbol{x}). \tag{160}$$

Thus, we get:

$$\int d\boldsymbol{r}\, d\boldsymbol{r}'\, \rho(\boldsymbol{r}, \boldsymbol{r}', \epsilon_f)\psi_n^*(\boldsymbol{r})\psi_n(\boldsymbol{r}') = \int d\boldsymbol{x}\, d\boldsymbol{x}'\psi_n^*(\boldsymbol{x})\psi_n(\boldsymbol{x}')\hat{C}(\boldsymbol{x}, \boldsymbol{x}', \omega)$$
$$\times \psi_i(\boldsymbol{x})\psi_i^*(\boldsymbol{x}')\delta(\epsilon_i - \epsilon_f - \omega). \tag{161}$$

Coming back to the definition of the density operator (12), one can write:

$$\rho(\boldsymbol{r}, \boldsymbol{r}', \epsilon_f) = \sum_m p_m \langle \boldsymbol{r}|\psi_m\rangle \langle \psi_m|\boldsymbol{r}'\rangle \delta(\epsilon_n - \epsilon_f). \tag{162}$$

Therefore, one can write:

$$\int d\boldsymbol{r}\, d\boldsymbol{r}'\, \rho(\boldsymbol{r}, \boldsymbol{r}', \epsilon_f)\psi_n^*(\boldsymbol{r})\psi_n(\boldsymbol{r}') = \int d\boldsymbol{r}\, d\boldsymbol{r}' \sum_m p_m \langle \boldsymbol{r}|\psi_m\rangle \langle \psi_m|\boldsymbol{r}'\rangle \langle \boldsymbol{r}'|\psi_n\rangle \langle \psi_n|\boldsymbol{r}\rangle \delta(\epsilon_n - \epsilon_f). \tag{163}$$

Using $\int d\boldsymbol{r}\, |\boldsymbol{r}\rangle \langle \boldsymbol{r}| = \text{Id}$, we get:

$$\int d\boldsymbol{r}\, d\boldsymbol{r}'\, \rho(\boldsymbol{r}, \boldsymbol{r}', \epsilon_f)\psi_n^*(\boldsymbol{r})\psi_n(\boldsymbol{r}') = \int d\boldsymbol{r} \sum_m p_m \langle \boldsymbol{r}|\psi_m\rangle \langle \psi_m|\psi_n\rangle \langle \psi_n|\boldsymbol{r}\rangle \delta(\epsilon_n - \epsilon_f), \tag{164}$$

which leads to:

$$\int d\boldsymbol{r}\, d\boldsymbol{r}'\, \rho(\boldsymbol{r}, \boldsymbol{r}', \epsilon_f)\psi_n^*(\boldsymbol{r})\psi_n(\boldsymbol{r}') = \int d\boldsymbol{r}\, p_n \langle \boldsymbol{r}|\psi_n\rangle \langle \psi_n|\boldsymbol{r}\rangle \delta(\epsilon_n - \epsilon_f). \tag{165}$$

Replacing the latest equation in (161), we obtain:

$$\int d\boldsymbol{r}\, p_n \langle \boldsymbol{r}|\psi_n\rangle \langle \psi_n|\boldsymbol{r}\rangle \delta(\epsilon_n - \epsilon_f) = \int d\boldsymbol{x}\, d\boldsymbol{x}'\, \psi_n^*(\boldsymbol{x})\psi_n(\boldsymbol{x}')\hat{C}(\boldsymbol{x}, \boldsymbol{x}', \omega)$$
$$\times \psi_i(\boldsymbol{x})\psi_i^*(\boldsymbol{x}')\delta(\epsilon_i - \epsilon_f - \omega). \tag{166}$$

Summing over $n$ we finally obtain:

$$\rho(\boldsymbol{r},\boldsymbol{r},\epsilon_f) = \sum_n \int d\boldsymbol{x}\, d\boldsymbol{x}'\, \psi_n^*(\boldsymbol{x})\psi_n(\boldsymbol{x}')\hat{C}(\boldsymbol{x},\boldsymbol{x}',\omega)\psi_i(\boldsymbol{x})\psi_i^*(\boldsymbol{x}')\delta(\epsilon_i-\epsilon_f-\omega). \quad (167)$$

Finally, observing that $\rho(\boldsymbol{r},\boldsymbol{r},\epsilon_f)$ is the probability of finding an electron at $\boldsymbol{r}$ with the energy $\epsilon_f$, one can directly identify the integral as the total electron energy loss probability $\Gamma(\omega)$ and get:

$$\Gamma(\omega) = \sum_n \int d\boldsymbol{x}\, d\boldsymbol{x}'\, \psi_n^*(\boldsymbol{x})\psi_n(\boldsymbol{x}')\hat{C}(\boldsymbol{x},\boldsymbol{x}',\omega).\psi_i(\boldsymbol{x})\psi_i^*(\boldsymbol{x}')\delta(\epsilon_i-\epsilon_f-\omega). \quad (168)$$

Replacing $\hat{C}$ by either its quasi-static or the retarded form, one respectively obtains Eq. (9) and (11).

## 7.2 Application to coherence measurements of optical fields

In the following, we will note $\boldsymbol{p}_f$ and $\boldsymbol{p}_i$ respectively the wave-vectors of the final and initial electrons. The subscript $z$ will denote the component of vectors parallel to the propagation axis while the subscript $\perp$ denotes the plane perpendicular to $z$. The vector $\boldsymbol{k}$ correspond to the conjugate variable of $\boldsymbol{r}$ therefore indexing the reciprocal space. First of all, let's calculate the Fourier transform of Eq. (155) in the plane $\perp$:

$$\rho_f\left(\boldsymbol{k}_\perp,\boldsymbol{k}'_\perp,r_z,r'_z\right) = \int d\boldsymbol{x}\, d\boldsymbol{x}'\, \mathscr{F}_{\boldsymbol{r}_\perp}\{U_0(\boldsymbol{r},\boldsymbol{x})\}\mathscr{F}_{-\boldsymbol{r}'_\perp}\{U_0^*(\boldsymbol{r},\boldsymbol{x})\}\hat{C}\left(\boldsymbol{x},\boldsymbol{x}',\omega\right)\rho_i\left(\boldsymbol{x},\boldsymbol{x}'\right), \quad (169)$$

where for brevity we omitted the energy in the argument of the density matrices. The free particle Green function reads [15]:

$$U_0(\boldsymbol{r},\boldsymbol{x}) = -\frac{1}{2\pi}\frac{e^{ip_f|\boldsymbol{r}-\boldsymbol{x}|}}{|\boldsymbol{r}-\boldsymbol{x}|}. \quad (170)$$

Therefore its Fourier transform is given by [15, 26]:

$$\mathscr{F}_{\boldsymbol{r}_\perp}\{U_0(\boldsymbol{r},\boldsymbol{x})\} = \frac{-i}{p_{f,z}}e^{-i\boldsymbol{k}_\perp.\boldsymbol{x}}e^{ip_{f,z}(r_z-x_z)} \quad (171)$$

and we moreover have $\mathscr{F}_{\boldsymbol{r}_\perp}\{U_0(\boldsymbol{r},\boldsymbol{x})\} = \mathscr{F}^*_{-\boldsymbol{r}'_\perp}\{U_0^*(\boldsymbol{r},\boldsymbol{x})\}$. The latter inserted in Eq. (169) gives:

$$\rho_f\left(\boldsymbol{k}_\perp,\boldsymbol{k}'_\perp,r_z,r'_z\right) = \frac{1}{p_{f,z}^2}e^{ip_{f,z}(r_z-r'_z)}\int d\boldsymbol{x}\, d\boldsymbol{x}'e^{-ip_{f,z}(x_z-x'_z)}e^{-\boldsymbol{k}_\perp.\boldsymbol{x}}e^{\boldsymbol{k}'_\perp.\boldsymbol{x}'}\hat{C}\left(\boldsymbol{x},\boldsymbol{x}',\omega\right)\rho_i\left(\boldsymbol{x},\boldsymbol{x}'\right). \quad (172)$$

Now the Fourier transforms with respect to $r_z$ and $r'_z$ become trivial and give:

$$\rho_f\left(\boldsymbol{k}_\perp,\boldsymbol{k}'_\perp,k_z,k'_z\right) = \frac{4\pi^2}{p_{f,z}^2}\delta(k_z-p_{f,z})\delta(k'_z-p_{f,z})\int d\boldsymbol{x}\, d\boldsymbol{x}'e^{-ip_{f,z}(x_z-x'_z)} \\ \times e^{-\boldsymbol{k}_\perp.\boldsymbol{x}}e^{\boldsymbol{k}'_\perp.\boldsymbol{x}'}\hat{C}\left(\boldsymbol{x},\boldsymbol{x}',\omega\right)\rho_i\left(\boldsymbol{x},\boldsymbol{x}'\right). \quad (173)$$

We can integrate over $k_z$ and $k'_z$ as they are not observed experimentally [15]:

$$\rho_f\left(\boldsymbol{k}_\perp,\boldsymbol{k}'_\perp\right) = \frac{4\pi^2}{v^2}\int d\boldsymbol{x}\, d\boldsymbol{x}'e^{-ip_{f,z}(x_z-x'_z)}e^{-\boldsymbol{k}_\perp.\boldsymbol{x}}e^{\boldsymbol{k}'_\perp.\boldsymbol{x}'}\hat{C}\left(\boldsymbol{x},\boldsymbol{x}',\omega\right)\rho_i\left(\boldsymbol{x},\boldsymbol{x}'\right), \quad (174)$$

where we used $p_{f,z} \approx mv/\hbar$. We now consider the case of the retarded interaction and make use of the paraxial approximation but in a slightly different formulation:

$$\rho_i\left(\boldsymbol{x}, \boldsymbol{x}'\right) = \frac{1}{L} \rho_{i,\perp}^e\left(\boldsymbol{x}_\perp, \boldsymbol{x}'_\perp\right) e^{ip_{i,z}x_z} e^{-ip_{i,z}x'_z}, \tag{175}$$

where $L$ denotes the interaction length between the probe electron and the sample. Moreover the incident electron kinetic energy being principally contained in its $z$-component, one can write [11]:

$$\boldsymbol{\nabla}\psi_i(\boldsymbol{r}) \approx \psi_i(\boldsymbol{r}) i k_i \hat{z} = \frac{imv}{\hbar} \psi_i(\boldsymbol{r}) \hat{z}. \tag{176}$$

Plugging Eq. (175), (176) and the retarded form of $K$ in (174), one gets:

$$\rho_f\left(\boldsymbol{k}_\perp, \boldsymbol{k}'_\perp\right) = \frac{4\pi^2}{L} \int d\boldsymbol{x}\, d\boldsymbol{x}' \mathrm{Im}\{-\mathcal{G}_{zz}(\boldsymbol{x}, \boldsymbol{x}', \omega)\} \rho_i\left(\boldsymbol{x}_\perp, \boldsymbol{x}'_\perp\right) e^{-i(p_{f,z}-p_{i,z})x_z}$$
$$\times e^{i(p_{f,z}-p_{i,z})x'_z} e^{-\boldsymbol{k}_\perp.\boldsymbol{x}} e^{\boldsymbol{k}'_\perp.\boldsymbol{x}'}. \tag{177}$$

The integration over $x_z$ and $x'_z$ gives:

$$\rho_f\left(\boldsymbol{k}_\perp, \boldsymbol{k}'_\perp\right) = \frac{4\pi^2}{L} \int d\boldsymbol{x}\, d\boldsymbol{x}' \mathrm{Im}\{-\mathcal{G}_{zz}(\boldsymbol{x}_\perp, \boldsymbol{x}'_\perp, \omega)\} \rho_i\left(\boldsymbol{x}_\perp, \boldsymbol{x}'_\perp, q, -q,\right) e^{-\boldsymbol{k}_\perp.\boldsymbol{x}} e^{\boldsymbol{k}'_\perp.\boldsymbol{x}'}. \tag{178}$$

Using the definition of the MCT (24), one can then conclude that:

$$\rho_f\left(\boldsymbol{k}_\perp, \boldsymbol{k}'_\perp\right) = \frac{2\pi^3}{\omega L} \int d\boldsymbol{x}\, d\boldsymbol{x}' \Lambda_{zz}(\boldsymbol{x}_\perp, \boldsymbol{x}'_\perp, q, -q, \omega) \rho_i\left(\boldsymbol{x}_\perp, \boldsymbol{x}'_\perp\right) e^{-\boldsymbol{k}_\perp.\boldsymbol{x}} e^{\boldsymbol{k}'_\perp.\boldsymbol{x}'}, \tag{179}$$

which simply reads:

$$\rho_f\left(\boldsymbol{k}_\perp, \boldsymbol{k}'_\perp\right) = \frac{2\pi^3}{\omega L} \Lambda_{zz}(\boldsymbol{k}_\perp, \boldsymbol{k}'_\perp, q, -q, \omega) * \rho_i\left(\boldsymbol{k}_\perp, \boldsymbol{k}'_\perp\right). \tag{180}$$

Finally, one can come back in the real space and deduce the rather elegant formula:

$$\rho_f\left(\boldsymbol{r}_\perp, \boldsymbol{r}'_\perp\right) = \frac{2\pi^3}{\omega L} \Lambda_{zz}(\boldsymbol{r}_\perp, \boldsymbol{r}'_\perp, q, -q, \omega) \rho_i\left(\boldsymbol{r}_\perp, \boldsymbol{r}'_\perp\right). \tag{181}$$

As we discussed earlier, Agarwal demonstrated [84], using the fluctuation-dissipation theorem, that the MCT is proportional to the electromagnetic correlation function. Thus, Eq. (181) shows that, when an electron is scattered by an optical field, *the electromagnetic correlations are imprinted in the coherence properties of the electron beam*. Producing electronic interferences thus constitutes a measurement of these correlations.

We can now connect Eq. (181) to the standard theory of electron holography. Indeed, during an inelastic interaction and for small scattering angles, the final and initial density matrices can be connected by the relation [59]:

$$\rho_f(\boldsymbol{r}_\perp, \boldsymbol{r}'_\perp, E - \omega) = T(\boldsymbol{r}_\perp, \boldsymbol{r}'_\perp, \omega) \rho_i(\boldsymbol{r}_\perp, \boldsymbol{r}'_\perp, E). \tag{182}$$

Here, $T(\boldsymbol{r}_\perp, \boldsymbol{r}'_\perp, \omega)$ denotes a general tensor which only depends on the scatterer and the energy loss $\hbar\omega$ and is usually referred to as the *mutual object transparency* (MOT). In 1985, Kohl and Rose demonstrated that, in the quasistatic limit, the MOT corresponds to the MDFF [14] but, so far no equivalent relation has been established for the retarded regime. Remarkably, Eq. (181) constitutes the extension to the retarded case of their results and rigorously demonstrates that in this case, the MOT corresponds to the Mutual coherence tensor.

The formalism recalled or developed here is the building block of inelastic electron holography. Such an experiment can be schematized in three steps:

1. We prepare an initial electron state which density matrix $\rho_i(r_\perp, r'_\perp, E)$ corresponds to a pure state. In standard off-axis electron holography, it simply corresponds to a plane-wave but, with modern phase-shaping techniques, it could corresponds to e.g. a vortex with a pure OAM.

2. The initial electron state is scattered by the sample to a set of final states. After an energy loss $\hbar\omega$ and for small scattering angles, the final density matrix is given by $\rho_f(E-\omega) = T(\omega)\,\rho_i(E)$ where the mutual object transparency corresponds: (1) to electronic charge correlation in the quasi-static regime or (2) to photon correlation in the retarded regime. In other words, the scattering event imprints the signature of the correlations in the target onto the beam density matrix. The final density matrix does not correspond to a pure state anymore but rather to mixed electron states i.e. a partially coherent electron beam [59]. The off-diagonal elements of the density matrix, which modulus gives the mutual coherence of the field [16], encodes the correlations in the scatterer.

3. We produce interferences in order to retrieve these off-diagonal elements and therefore obtain information on the electronic or photonic correlations in the target.

Our formalism thus paves the road toward electron holography of optical field as preliminary investigated in the case of surface plasmon in e.g. [96].

# 8 Conclusion and perspectives

In this work, we have established a fully retarded formalism of fast electron inelastic scattering which can be used to described any type of TEM spectroscopy experiments such as low-loss and core-loss EELS, inelastic holography or energy-filtered 4D-STEM. Our work is built upon general response tensors including both quantum and relativistic aspects. Also, we made no assumptions on the peculiar details of the sample under consideration. Consequently, our formalism can be applied to a large set of systems and combined with any numerical methods from ab-initio to classical electrodynamics simulations. The core of our work relies on the introduction of a relativistic extension of the celebrated mixed dynamic form factor as the Fourier transform of the 4-susceptibility. By connecting this new quantity to the mutual coherence tensor - the central object of the theory of optical fields - we have drawn a formal and rigorous connection between the condensed-matter and nano-optical approaches, thus encompassing all the existing theoretical developments for EELS in an unique and general framework. Then, taking a careful consideration of most of the possible approximations, we have demonstrated that most of the approaches generally employed in the literature can be deduced from our formalism.

Beyond this effort of synthesis and unification, we believe that our work paves the road toward new experiments and interpretations. First of all, as thoroughly discussed in Sec. 7.2, by introducing a relativistic form of the kinetic equation, our work enables to take into account retardation effects in holographic experiments. This constitutes the key ingredient to design and model new experiments enabling the measurement of the cross density of states [69, 70] directly at the nano-scale, following an original idea of García de Abajo [11]. Moreover, our work now enables the application of all the powerful tools developed for electron hologra-

phy [59, 97] to the nano-photonic domain. Particularly, recent developments in differential phase contrast or ptychography for plasmonics should be described with this language. Secondly, our work bridging nano-optical and condensed-matter formalisms, we foresee a mutual and beneficial exchange of concepts between these two approaches. On the one hand, introducing the retarded screened interaction to solid state systems, one could interpret core-loss spectroscopy experiments in terms of X-ray photon exchange. This would give a natural interpretation of the already known mathematical close identity between EELS and inelastic X-ray scattering [98]. Such a comparison would be exactly similar to the standard analogy between EELS and optical extinction experiments on surface plasmons [99]. On the other hand, employing the relativistic MDFF to describe low-loss spectroscopy experiments directly paves the way toward the ab-initio modelling of EELS experiments on nano-optical systems, with potential and far-reaching applications in, e.g., quantum plasmonics [100] or exciton-plasmon [101, 102] and phonon-plasmon coupling physics [101, 103]. Also, this should ease bridging the antagonist descriptions of EELS phonon spectroscopy experiments, either quantum [104–107] or classical, in the quasi-static [108, 109] or retarded [110] regimes, or attempts to mix the two [111]. Besides, by putting the relativistic MDFF at the core of our approach, we enable the extensive use of ab-initio methods for the modelling of EELS experiment, in the same vein as the pioneering works employing density functional theory [112,113]. Finally, the introduction of the 4-current correlation function enables to employ the time-dependent current-density-functional theory [114] to model TEM spectroscopy experiments, which has never been done so far to the best of our knowledge. Such an approach would be particularly well suited to model e.g. EMCD experiments or relativistic effects in core-loss EELS.

# Acknowledgement

AL acknowledges funding from the European Research Council (ERC) under the Horizon 2020 research and innovation program of the European Union (grant agreement number 715620). This project has received funding from the European Unions horizon 2020 research and innovation programme under grant agreement No 823717.

# A Conventions, units, notations and Green functions

## A.1 Conventions and notations

The metric $g_{\mu,\nu}$ for the Minkowski space $\mathbb{M}^4$ is chosen with the signature $(+,-,-,-)$ i.e.

$$g_{\mu,\nu} = g^{\mu,\nu} = \begin{pmatrix} 1 & 0 & 0 & 0 \\ 0 & -1 & 0 & 0 \\ 0 & 0 & -1 & 0 \\ 0 & 0 & 0 & -1 \end{pmatrix}.$$

Under this convention, the raising or lowering of a spatial index changes the sign of a tensor; raising or lowering the temporal index leaves the sign unchanged. Unless otherwise specified, we have always used in this paper the implicit Einstein summation on repeated indices:

$$x^\mu x'_\mu \equiv \sum_{\mu,\nu=0}^{4} g^{\mu,\nu} x^\mu x'_\nu = c^2 t t' - \boldsymbol{x}.\boldsymbol{x}'. \tag{183}$$

The Fourier transform in $\mathbb{M}^4$ is defined as:

$$\begin{cases} f(\mathrm{x}) = \displaystyle\int_{\mathbb{M}^4} \frac{d^4\mathrm{k}}{(2\pi)^4} f(\mathrm{k}) \, e^{ik_\mu \mathrm{x}^\mu} & (184\mathrm{a}) \\[4mm] f(\mathrm{k}) = \displaystyle\int_{\mathbb{M}^4} d^4\mathrm{x} \, f(\mathrm{x}) \, e^{-ik_\mu \mathrm{x}^\mu} & (184\mathrm{b}) \end{cases}$$

where the 4-wavevector is defined as $k^\mu = (\omega/c, \boldsymbol{k})$. We define the 4-gradient as:

$$\partial_\mu = \frac{\partial}{\partial \mathrm{x}^\mu} = \left( \frac{1}{c} \frac{\partial}{\partial t}, \boldsymbol{\nabla} \right). \tag{185}$$

We can therefore define the 4-impulsion operator:

$$p_\mu = i\hbar \partial_\mu = \left( \frac{i\hbar}{c} \frac{\partial}{\partial t}, i\hbar \boldsymbol{\nabla} \right) \tag{186}$$

and in presence of a EM field, one needs to perform the minimal substitution $p_\mu \to p_\mu - qA_\mu$, $q$ being the charge of the particle. Moreover, the 4-current associated with a wavefunction $\psi$ can be expressed as:

$$j_\mu = i \left( \psi^* \partial_\mu \psi - \psi \partial_\mu \psi^* \right). \tag{187}$$

## A.2 Correlators and Green functions

The time ordering operator $\mathcal{T}$ between two fields $A(\mathrm{x})$ and $B(\mathrm{y})$ is defined as:

$$\begin{aligned} \mathcal{T}\{A(\boldsymbol{r},t)B(\boldsymbol{r}',t')\} = {}& \theta(t-t')A(\boldsymbol{r},t)B(\boldsymbol{r}',t') \\ & \pm \theta(t'-t)B(\boldsymbol{r}',t')A(\boldsymbol{r},t), \end{aligned} \tag{188}$$

where a $+$ sign applies for bosons and a $-$ sign for fermions. For a scalar field $A$, one can also define two different Green functions:
• The retarded Green function:

$$G^{\mathrm{R}}(\boldsymbol{r},\boldsymbol{r}',t,t') = -i\theta(t-t') \left\langle [A(\boldsymbol{r},t),A(\boldsymbol{r}',t')]_\pm \right\rangle_0. \tag{189}$$

• The causal Green function:

$$G^{\mathrm{C}}(\boldsymbol{r},\boldsymbol{r}',t,t') = -i \left\langle \mathcal{T}\{A(\boldsymbol{r},t)A(\boldsymbol{r}',t')\} \right\rangle_0. \tag{190}$$

In each case, $\langle . \rangle$ represents the statistical average value at thermal equilibrium and $[,]_\pm$ represents the fermion anti-correlator (resp. boson correlator).

## A.3 Lagrangian form of the Maxwell equations

The four-potential defined as $\mathrm{A}^\nu = (\phi/c, \boldsymbol{A})$ and the four-current defined as $\mathrm{J}^\nu = (c\rho, \boldsymbol{j})$ are connected by the equation of motion for the EM field:

$$\partial^\nu \partial_\mu \mathrm{A}^\mu - \partial^\mu \partial_\mu \mathrm{A}^\nu = -4\pi \mathrm{J}^\nu, \tag{191}$$

where $A_\mu$ is defined up to a scalar gauge function $\Lambda$:

$$\mathrm{A}_\mu(\mathrm{x}) \longrightarrow \mathrm{A}_\mu(\mathrm{x}) + \partial_\mu \Lambda(\mathrm{x}). \tag{192}$$

The anti-symmetric Faraday tensor $F^{\mu\nu}$ is defined as:

$$\mathrm{F}^{\mu\nu} = \partial^\mu \mathrm{A}^\nu + \partial^\nu \mathrm{A}^\mu, \tag{193}$$

which explicitly reads:

$$F^{\mu\nu} = \begin{pmatrix} 0 & -E_x & -E_y & -E_z \\ E_x & 0 & -B_z & B_y \\ E_y & B_z & 0 & -B_x \\ E_z & -B_y & B_x & 0 \end{pmatrix}. \tag{194}$$

For any anti-symmetric tensor $T$, we also introduce the *Hodge dual* as:

$$^\star T^{\alpha\beta} = \frac{1}{2}\epsilon^{\alpha\beta\mu\nu}T_{\mu\nu}, \tag{195}$$

where $\epsilon^{\alpha\beta\mu\nu}$ is the Levi-Civita pseudotensor defined as:

$$\epsilon^{\alpha\beta\mu\nu} = \begin{cases} +1, & \text{for an even permutation of } (0,1,2,3) \\ -1, & \text{for an odd permutation of } (0,1,2,3) \\ 0, & \text{otherwise} \end{cases}. \tag{196}$$

The Maxwell equations then read:

$$\begin{cases} \partial_\mu F^{\mu\nu} = J^\nu & \text{(197a)} \\ \partial_\mu (^\star F^{\mu\nu}) = 0 & \text{(197b)} \end{cases}$$

The last equation can be derived from the Lagrange equation applied to the *standard* EM Lagrangian density defined as $\mathcal{L}$:

$$\mathcal{L} = -\frac{1}{4}\left(\partial_\alpha A_\beta - \partial_\beta A_\alpha\right)\left(\partial^\alpha A^\beta - \partial^\beta A^\alpha\right) - J_\alpha A^\alpha. \tag{198}$$

The first term concerns only the EM field while the second is the field-source interaction.

# B  Reciprocity theorem and symmetry properties of the Green dyadic

The reciprocity theorem corresponds to the following condition:

$$\mathscr{S}\left(\mathcal{G}_j^i(\mathbf{r}',\mathbf{r},\omega)\right) = \mathcal{G}_j^i(\mathbf{r}',\mathbf{r},\omega), \tag{199}$$

where the operator $\mathscr{S}$ exchanges the indexes $i \leftrightarrow j$ and coordinates $\mathbf{r} \leftrightarrow \mathbf{r}'$. In this section, we examine the symmetry of the four tensors involved in the definition of $G$ (89) by the application of $\mathscr{S}$. We first remind that (at least in the three gauges considered in 3.1), the vacuum photon propagators satisfy the property:

$$D^{i0} = 0 \tag{200}$$

In other words, the temporal and the spatial components of the EM fields are not coupled in vacuum. Moreover, we recall the definition of the retarded screened interaction (73):

$$\mathcal{D}_\alpha^\beta(\mathbf{r}',\mathbf{r},\omega) = \int d\mathbf{r}_1 d\mathbf{r}_2\, D_\mu^\beta(\mathbf{r}',\mathbf{r}_2)\,\chi_\nu^\mu(\mathbf{r}_2,\mathbf{r}_1,\omega)\,D_\alpha^\gamma(\mathbf{r}_1,\mathbf{r}). \tag{201}$$

Let's consider the first term of Eq. (89) and examine its symmetry. The relation (200) leads to:

$$\mathcal{D}_0^0(\mathbf{r},\mathbf{r}') = \int d\mathbf{r}_1 d\mathbf{r}_2\, D_0^0(\mathbf{r},\mathbf{r}_2)\,\chi_0^0(\mathbf{r}_2,\mathbf{r}_1)\,D_0^0(\mathbf{r}_1,\mathbf{r}'). \tag{202}$$

The vacuum photon propagators can be straightforwardly reversed as $D_0^0(r, r_2) = D_0^0(r_2, r)$ and $D_0^0(r_1, r') = D_0^0(r', r_1)$. From Eq. (69), the charge part can then be written:

$$\chi_0^0(r_2, r_1, \omega) = 2 \sum_n \langle 0| J^0(r_2) |n\rangle \langle n| J_0(r_1) |0\rangle \Theta(\omega + \omega_n - \omega_0), \tag{203}$$

where $\Theta(\omega) = \frac{1}{\omega + i\eta}$. One can then see that $\chi_0^0(r_2, r_1) = (\chi_0^0(r_1, r_2))^\dagger$ because the lowering and raising of 0 indexes won't bring any sign changes. Finally, we notice that $\nabla^j \nabla'_i = \nabla'^i \nabla_j$ because the raising of $i$ and the lowering of $j$ will give both a minus sign. Indeed, $\nabla^j = \delta_i^j g^{ij} \nabla_j = -\nabla_j$ because $g^{jj} = -1$ by definition of the metric we have chosen. Thus, we finally have:

$$\mathscr{S}\left(\nabla'^i \nabla_j \mathcal{D}_0^0(r', r)\right) = \nabla'^i \nabla_j \mathcal{D}_0^0(r', r). \tag{204}$$

The same arguments leads to[2]:

$$\mathscr{S}\left(\mathcal{D}_j^i(r', r)\right) = \mathcal{D}_j^i(r', r). \tag{205}$$

We finally need to look at the last part of $G$ i.e.:

$$M_j^i(r', r) \equiv \partial_j \mathcal{D}_0^i(r', r) + \partial'^i \mathcal{D}_j^0(r', r). \tag{206}$$

We thus calculate:

$$\mathscr{S}\left(M_j^i(r', r)\right) = \nabla'_i \mathcal{D}_0^j(r, r') + \nabla^j \mathcal{D}_i^0(r, r') \tag{207}$$

$$= -\nabla'^i \mathcal{D}_0^j(r, r') - \nabla_j \mathcal{D}_i^0(r, r'). \tag{208}$$

Moreover, the first photon propagator reads:

$$\mathcal{D}_0^j(r, r') = \int dr_1 dr_2 D_0^0(r, r_2) \chi_a^0(r_2, r_1) D_a^j(r_1, r'), \tag{209}$$

where we used Eq. (200). We can again reverse the vacuum photon propagators which are obviously symmetric. However, the susceptibility term is *antisymmetric*. Indeed, it corresponds to a charge-current correlator and lowering the time part will keep the sign unchanged, while raising the spatial part will give a minus sign. Therefore:

$$\mathcal{D}_0^j(r, r') = -\mathcal{D}_j^0(r', r). \tag{210}$$

And similarly:

$$\mathcal{D}_i^0(r, r') = -\mathcal{D}_0^i(r', r). \tag{211}$$

We therefore finally have:

$$\mathscr{S}\left(M_j^i(r', r)\right) = -\nabla'^i \mathcal{D}_j^0(r', r) - \nabla_j \mathcal{D}_0^i(r', r) \tag{212}$$

$$= -M_j^i(r', r). \tag{213}$$

The $M$ tensor is therefore antisymmetric. To guarantee the symmetry of $G$, we must have:

$$M_j^i(r', r) = 0 \tag{214}$$

Therefore, in a *reciprocal medium*, the Green dyadic reads:

$$\mathcal{G}_j^i(r', r, \omega) = -\frac{1}{4\pi\omega^2} \nabla'^i \nabla_j \mathcal{D}_0^0(r', r, \omega) + \frac{1}{4\pi c^2} \mathcal{D}_j^i(r', r, \omega), \tag{215}$$

where the first term is a charge-charge correlator while the second term is a current-current correlator.

---

[2]The only difference in this case is that the raising and lowering of indices in the electron part will give two minus signs which cancel out.

# C Relativistic anisotropy in core-loss scattering

In the following lines we will further approximate the MDFF eventually ariving at a simplified description, which is useful in the core-loss regime. The final result has been previously employed to explain the mismatch between experimentally measured and non-relativistically predicted magic scattering angles. As stated in the main text, the transition probability obtained from the Feynman diagram depicted in Fig. 4(b) can be rewritten as a spatial integral inserting beam energy eigenstates (cf. Eq. (47))

$$\Gamma_{i\to f} = 2\pi \int d\mathbf{r}\, d\mathbf{r}'\, J_{fi}^{\mu}(\mathbf{r})\, J_{fi}^{\nu*}(\mathbf{r}')\, \mathscr{F}_{\mathbf{r},-\mathbf{r}'}^{-1}\left\{\frac{\mathcal{S}_{\mu\nu}(\mathbf{k},\mathbf{k}',\omega)}{(k^2-\omega^2)(k'^2-\omega^2)}\right\}, \tag{216}$$

with the relativistically generalized MDFF

$$S_{\mu\nu}(\mathbf{k},\mathbf{k}',\omega) = 2\pi \sum_f \langle i|\mathbf{j}_\mu(\mathbf{k})|f\rangle \langle f|\mathbf{j}_\nu^\dagger(\mathbf{k}')|i\rangle\, \delta(\omega+\omega_f-\omega_i). \tag{217}$$

We first note that the time-component of the transition current can be explicitely evaluated to

$$J^0(\mathbf{r}) = (E_f+E_i)\psi_f^*(\mathbf{r})\psi_i(\mathbf{r}) \approx 2E_i\psi_f^*(\mathbf{r})\psi_i(\mathbf{r}). \tag{218}$$

In a next step we take into account that (in)elastic scattering in the TEM is concentrated around the incident beam direction $z$ (paraxial regime). As a consequence, we may neglect all terms containing current components in $x-$ and $y-$ direction. Additionally the $z-$component can be approximated by

$$J^3(\mathbf{r}) \approx (k_f+k_i)\psi_f^*(\mathbf{r})\psi_i(\mathbf{r}) \approx 2k_i\psi_f^*(\mathbf{r})\psi_i(\mathbf{r}). \tag{219}$$

The above approximations are generally valid for TEM-EELS (i.e., employing fast electrons and considering energy losses well below the beam electron energy). The following two approximations to $S$ more specifically apply to the core-loss regime. We first consider non-relativistic target atomic wave functions $\xi$, which are not subject to spin-orbit coupling or other relativistic corrections. In that particular case we can approximate the time-component of the (relativistic) target transition current in the general expression (217) with the electron rest mass:

$$\langle i|\mathbf{j}_0(\mathbf{k})|f\rangle \approx 2m\langle i|e^{i\mathbf{k}\mathbf{r}}|f\rangle. \tag{220}$$

Moreover, we can employ the (non-relativistic) commutator $\mathrm{p} = im[\hat{H},\mathrm{r}]$ to replace the $p_z$ operator in the $z-$ component of the transition current yielding

$$\langle i|\mathbf{j}_3(\mathbf{k})|f\rangle = im\omega\langle i|z e^{i\mathbf{k}\mathbf{r}}|f\rangle. \tag{221}$$

Last but not least we use the dipole approximation to simplify $e^{i\mathbf{q}\mathbf{r}} \approx 1+i\mathbf{q}\mathbf{r}$ and $z e^{i\mathbf{q}\mathbf{r}} \approx z$, i.e., only linear terms in spatial coordinates are kept in the strongly localized integrals of the atomic wave functions. Inserting all these approximations into (216) and (217) we obtain:

$$\Gamma_{i\to f} = 2\pi \int d\mathbf{r}\, d\mathbf{r}'\, \psi_f^*(\mathbf{r})\psi_i(\mathbf{r})\psi_f(\mathbf{r}')\psi_i^*(\mathbf{r}')\, \mathscr{F}_{\mathbf{r},-\mathbf{r}'}^{-1}\left\{\frac{\mathcal{S}^{\mathrm{R}}(\mathbf{k},\mathbf{k}',\omega)}{(k^2-\omega^2)(k'^2-\omega^2)}\right\}, \tag{222}$$

with:

$$S^{\mathrm{R}}(\mathbf{k},\mathbf{k}',\omega) = S^{\mathrm{QS}}(\mathbf{k}_\perp,\mathbf{k}'_\perp, k_z-k_i\omega/E_i, k'_z-k_i\omega/E_i, \omega.) \tag{223}$$

Accordingly, we obtained a simplified approximation for the retarded loss probability in the core loss regime, which basically consists of rescaling the arguments in the non-retarded expression. Notably, the correction factor for the $z$-momentum (i.e., scattering angle) in the MDFF reads

$$\frac{k_i \omega}{E_i} = \frac{\gamma m_e v_i \omega}{\gamma m_e} = v_i \omega, \tag{224}$$

($= v_i \omega/c^2$ in SI units) which is exactly the correction of the $z-$momentum obtained in Refs. [74–76]. The above MDFF (eventually including further approximations) is frequently used in EELS computations of core losses.

# D    The self-energy approach

In 1987, Echenique and collaborators demonstrated Eq. (9) using a different approach based on the calculation of the probe electron self-energy [54]. Their formalism have the advantage to be compact and easily applicable although they did not provide details of the demonstration in their paper. Here, we briefly demonstrate that their equation can be formally extracted from our latter developments. In Sec. (6.2), we calculated the self energy $\hat{\Sigma}$ of the electron and obtained:

$$\hat{\Sigma}(\boldsymbol{r}, \boldsymbol{r}', t, t') = \hat{U}_0(\boldsymbol{r}, \boldsymbol{r}', t, t')\hat{C}(\boldsymbol{r}, \boldsymbol{r}', t, t'), \tag{225}$$

where we recall that $\hat{C}(\boldsymbol{r}, \boldsymbol{r}', t, t') = \langle \mathcal{T}\{\hat{V}(\boldsymbol{r}, t)\hat{V}(\boldsymbol{r}, t)\}\rangle$. Since all the quantities above only depend on $t - t'$, Fourier transform the latter expression will give the following convolution product:

$$\hat{\Sigma}(\boldsymbol{r}, \boldsymbol{r}', E) = \int d\omega \, \hat{U}_0(\boldsymbol{r}, \boldsymbol{r}', E + \omega)\hat{C}(\boldsymbol{r}, \boldsymbol{r}', -\omega). \tag{226}$$

Moreover, the Fourier transform of the electron propagator is simply [26]:

$$\hat{U}_0(\boldsymbol{r}, \boldsymbol{r}', E + \omega) = \frac{1}{E + \omega - \hat{H}_e + i0^+}. \tag{227}$$

Therefore, the self-energy in the spectral domain reads:

$$\hat{\Sigma}(\boldsymbol{r}, \boldsymbol{r}', E) = \int d\omega \, \frac{\hat{C}(\boldsymbol{r}, \boldsymbol{r}', -\omega)}{E + \omega - \hat{H}_e + i0^+}. \tag{228}$$

The mean energy $\Sigma_0$ of an electron of wavefunction $|\psi_0\rangle$ and energy $E_0$ is:

$$\Sigma_0 = \langle \psi_0 | \hat{\Sigma} | \psi_0 \rangle. \tag{229}$$

Inserting the completeness relation $\sum_f |\psi_f\rangle \langle \psi_f| = 1$ for a basis of final states and two others for the $\{|r\rangle\}$ and $\{|r'\rangle\}$ basis, we obtain:

$$\Sigma_0 = \sum_f \int d\boldsymbol{r} \, d\boldsymbol{r}' \, \frac{\psi_0(\boldsymbol{r})\psi_0^*(\boldsymbol{r}')\hat{C}(\boldsymbol{r}, \boldsymbol{r}', -\omega)\psi_f(\boldsymbol{r}')\psi_f^*(\boldsymbol{r})}{E + \omega - E_0 + i0^+}. \tag{230}$$

Replacing $\hat{C}$ by its quasi-static form, we obtain the equations (3) of Echenique et al. [54]. Moreover, if we replace $\hat{C}$ by its retarded form, we obtain the retarded form of the self-energy formalism of Echenique et al.

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
