# Peer review of "Bridging nano-optics and condensed matter formalisms in a unified description of inelastic scattering of relativistic electron beams"

_SciPost Physics, doi:SciPost Phys. 10, 031 (2021)_

## Round 2 · Referee Report · Jo Verbeeck (Referee 1) · 2020-9-17

Strengths

  1. Develops common foundation for EELS and nanoptical experiments
  2. Builds on many earlier theoretical developments but puts them under one unifying umbrella
  3. Will form the basis for interpreting many new EELS related experiments involving phase manipulation, phonon scattering, time modulated beams and many more.
  4. Very thorough development of the theory, giving attention to all assumptions applied

Weaknesses

  1. Can be heavy to digest for the experimental reader

Report

This manuscript describes a theoretical derivation of the foundation of inelastic energy loss spectroscopy in a transmission electron microscope and its link to nano optics.
I would describe the manuscript as nothing less than a master piece, bringing together many of the concepts and theories that underpin all experimental work in this domain. The paper develops in a pedagogical way, a common description on how accelerated electron beams inelastically interact with a sample taking into account both quantum and relativistic aspects, augmenting and clearing up the work of previous derivations.
The authors succeed in giving a fair overview of the existing state of the art, while pointing out how these fit together in a more generalised scheme.
The findings are very important in a time where more control over the quantum state of the electron in such experiments is becoming available and where confusion exists in the community as to how to interpret its results in terms of the materials properties.
The results give hope that much more is to be expected from novel electron spectroscopic setups, also going beyond the assumptions made in this manuscript (time modulated beams, phonons, ...).

Requested changes

Minor comments:

  1. p1: "The overlap of the electron beam with the sample e.g ...". It is not clear what is meant with 'overlap' here. I think it refers to aloof experiments vs. bulk? Otherwise both suface and bulk effects play a role and seem to be independent on the overlap of the beam with the sample?
  2. fig1: an ending bracket is missing in the description of the illumination system
  3. p2: the magic angle is defined as the angle at which electrons are most likely deflected. This is confusing to me, but may be right. I would define the magic angle as the collection angle at which there is no dependence of the recorded spectrum on the orientation of an anisotropic sample.
  4. p2: "X or optical ...", I would write "X-ray or optical photon"
  5. p2: "in a STEM has yet been provided, which includes...", I would write "in a STEM has been provided, which implies..."
  6. p3: "the later to the retarded case,...", I think this needs to be "latter". I suggest to search for the word "later" and replace with "latter" wherever appropriate as this occurs several times in the manuscript.
  7. ref 18: check the spelling of Hebert (accents)
  8. fig2: the 'grey' part of the figure is not easy to distinguish
  9. p5: "The zz indices hind at", I think this needs to be "hint"
  10. p7: "The are therefore not sufficient to model such experiments". I would argue that even for 'simple' EELS experiments, the partial coherence of the outgoing electron wave does make an important difference as even going to diffraction space will make such coherence visible in the recorded intensities. In this sense, I would argue that we always need to keep the formalism developed in this manuscript in mind, even for the seemingly simpler setups.
  11. p7: "In other words, non-zero out of diagonal terms entail electron interferences in the image plane". Well, this depends on which plane you are looking at and in which plane the density matrix was given. E.g. if the image plane would be in r-space, the intensity would only depend on the diagonal (assuming no lens effects etc.).
  12. p7: "Upon comparision"...should be "comparison"
  13. p9: "Thanks to the Schwarz's inequality...", no need for the "'s"
  14. p9: "For the sake of clearness", could be "for the sake of clarity"
  15. p11: Around e.q. 54, the temperature is assumed zero. I wonder if the equations are linear whether the effect of the ignored temperature can later be added as an additive effect on top of the beam driven effect described here?
  16. p15: "at it generalize the Kubo approach...", should be "generalises"
  17. p15: after caption on concluding remarks "one usually calculate the Green dyadic" should be "calculates"
  18. p16: after eq. 105 "This term no encompass elastic...", should be "encompasses"
  19. p18: after eq. 119 "for a set of Gaussian random variable", should be "variables"
  20. p18: above eq. 123 "simulataneous" should be "simultaneous"
  21. p22: after caption VII, "We will therefore make furthers approximations", should be "further"
  22. p23: after eq. 169: "one respectively obtain equations (10)...", should be "obtains"

---

## Round 3 · Referee Report · Anonymous (Referee 2) · 2020-12-9

Strengths

  1. conceptual clarity,
  2. manuscript complete and self-contained
  3. presents unified theory for EELS
  4. connects directly to experimental method

Report

The authors present a unified theoretical description of inelastic electron scattering as relevant for modern day electron energy electron loss experiments in transmission electron microscopes. The manuscript is of exceptional conceptual and methodological clarity. The theoretical treatment and methods are of direct relevance to experimentalists in the field.

---

## Round 3 · Referee Report · Jo Verbeeck (Referee 1) · 2020-12-9

Report

The authors have carefully adressed all points and I considder the paper ready for publication as is.

---

## Round 3 · Author Response

We thank the referee for his thorough reading of the manuscript and for his very positive comments. We have considered all the points raised in the review. In details:

  1. p1: "The overlap of the electron beam with the sample e.g ...". It is not clear what is meant with 'overlap' here. I think it refers to aloof experiments vs. bulk? Otherwise both suface and bulk effects play a role and seem to be independent on the overlap of the beam with the sample?

We thank the referee for raising this ambiguity. We indeed refer to the distinction between aloof and penetrating trajectories. To clarify this point, we replaced this sentence by: “The spatial overlap of the electron beam with the sample, e.g., the importance of bulk versus surface effects.”

  1. fig1: an ending bracket is missing in the description of the illumination system

We thank the referee for having carefully checked the formatting of the figures. Nonetheless, we did not find the above-mentioned typo.

  1. p2: the magic angle is defined as the angle at which electrons are most likely deflected. This is confusing to me, but may be right. I would define the magic angle as the collection angle at which there is no dependence of the recorded spectrum on the orientation of an anisotropic sample.

The referee is right, and we have therefore modified our definition according to his suggestion. The new text reads: “For example, it has been demonstrated that the so-called magic angle, at which the dependence of the core-loss electron scattering on the orientation of an anisotropic sample is canceled, strongly depends on the retarded character of the electron to target interaction, which had been considered as negligible in core-loss investigations before.”

  1. p2: "X or optical ...", I would write "X-ray or optical photon"
  2. p2: "in a STEM has yet been provided, which includes...", I would write "in a STEM has been provided, which implies..."
  3. p3: "the later to the retarded case,...", I think this needs to be "latter". I suggest to search for the word "later" and replace with "latter" wherever appropriate as this occurs several times in the manuscript.

Points 4, 5 and 6 have been addressed and modified as the reviewer suggested.

  1. ref 18: check the spelling of Hebert (accents)

After checking, it appears that the spelling used in this manuscript is correct.

  1. fig2: the 'grey' part of the figure is not easy to distinguish

The figure has been enlarged and the gray adjusted to remedy that problem.

  1. p5: "The zz indices hind at", I think this needs to be "hint"

This typo has been correct.

  1. p7: "The are therefore not sufficient to model such experiments". I would argue that even for 'simple' EELS experiments, the partial coherence of the outgoing electron wave does make an important difference as even going to diffraction space will make such coherence visible in the recorded intensities. In this sense, I would argue that we always need to keep the formalism developed in this manuscript in mind, even for the seemingly simpler setups.

We have taken the remark of the referee into account and modified the text accordingly: “They are therefore not sufficient to generally model EELS experiments in the TEM and represent certain limiting cases where the above effects may be neglected”.

  1. p7: "In other words, non-zero out of diagonal terms entail electron interferences in the image plane". Well, this depends on which plane you are looking at and in which plane the density matrix was given. E.g. if the image plane would be in r-space, the intensity would only depend on the diagonal (assuming no lens effects etc.).

The referee is correct. We changed the pertaining phrase to: “In other words, non-zero off-diagonal terms entail electron interferences in the particular plane considered (which is defined by $z$ coordinate along the optical axis).”

  1. p7: "Upon comparision"...should be "comparison"
  2. p9: "Thanks to the Schwarz's inequality...", no need for the "'s"
  3. p9: "For the sake of clearness", could be "for the sake of clarity"

Points 12, 13 and 14 have been addressed and modified as the reviewer suggested.

  1. p11: Around e.q. 54, the temperature is assumed zero. I wonder if the equations are linear whether the effect of the ignored temperature can later be added as an additive effect on top of the beam driven effect described here? Finite temperature effects can be included in the linear response theory through the use of e.g. Matsubara Green functions. Thus, such effects can be added to our theory a posteriori through a simple re-definition of the propagators (e.g. the MDFF).

  2. p15: "at it generalize the Kubo approach...", should be "generalises"

  3. p15: after caption on concluding remarks "one usually calculate the Green dyadic" should be "calculates"
  4. p16: after eq. 105 "This term no encompass elastic...", should be "encompasses"
  5. p18: after eq. 119 "for a set of Gaussian random variable", should be "variables"
  6. p18: above eq. 123 "simulataneous" should be "simultaneous"
  7. p22: after caption VII, "We will therefore make furthers approximations", should be "further"
  8. p23: after eq. 169: "one respectively obtain equations (10)...", should be "obtains"

Points 16 to 22 have been addressed and modified as the reviewer suggested.

---

## Editorial Decision

published